# ANYCAP: OMNI-MODAL CAPTIONING WITH INSTRUCTION ALIGNMENT

## ABSTRACT

We present AnyCap, a plug-and-play framework that brings instruction alignment to omni-modal captioning. Captions offer a unified language interface for multimodal learning, and users increasingly expect instruction-driven control over their content and style. Current caption models lack explicit instruction supervision and are weak at instruction following, while directly tuning them can degrade general language ability. Achieving instruction alignment in an omni-modal setting is harder still, as each modality calls for separate models and custom designs. To address these challenges, AnyCap leverages a residual-correction paradigm that refines uncontrolled captions from existing models to instruction-aligned ones, without re-training base models. By processing multi-modality features in a unified framework, it enables one model to serve images, videos, and audio. To address the lack of instruction-based data, we construct AnyCapData, a large-scale, high-quality corpus spanning three modalities with 28 well-designed instruction types. For evaluation, we address the limitations of current metrics for instruction-oriented captioning by designing AnyCapEval. Its key insight is to decouple evaluation into content and style for fine-grained assessment. Extensive experiments show that on AnyCapEval and diverse public benchmarks, AnyCap consistently improves both caption quality and instruction adherence for both open-source and API-based models. Notably, AnyCap-8B boosts GPT-4o's content scores by $46\%$ and style scores by $12\%$. Our code and models will be made publicly available.

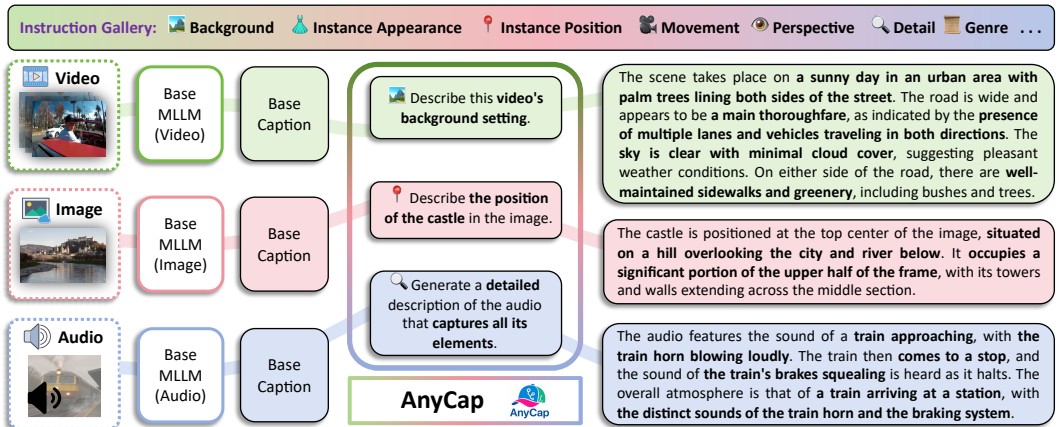

Figure 1: We design diverse natural language instructions for controllable captioning. Given image, video, and audio inputs, base models first generate initial captions. AnyCap effectively aligns them into high-quality, instruction-aligned outputs. A single AnyCap model serves all three modalities.

## 1 INTRODUCTION

Multimodal large language models (MLLMs) are rapidly advancing towards omni-modal intelligence (Hurst et al., 2024; Anil et al., 2023; Xu et al., 2025; Zhao et al., 2025). Textual captions provide a natural way to connect different modalities by conveying their information in a unified language form. Beyond producing generic descriptions, users increasingly expect captions that are

instruction-aligned, directly adapting to desired content and style. This can more flexibly support multimodal tasks such as retrieval (Radford et al., 2021), question answering (Li et al., 2022), and content generation (Ramesh et al., 2022).

Yet, instruction-aligned captioning faces three significant challenges. *(i)* Existing models have limited instruction following ability. Many are trained as general-purpose understanding models. They can lean toward the requested focus, but mainly produce broad content or unrelated details (Wang et al., 2025). For finer control over captions, current approaches usually rely on special tokens (Dwibedi et al., 2024) or bounding boxes (Zhao et al., 2024; Huang et al., 2024), to guide local descriptions. In practice, however, users could prefer flexible natural language instructions (*e.g.*, in Table 1) to specify caption types, focuses, and styles. This represents a more fine-grained and natural form of control, yet current captioning models fall short in this ability. Bridging the gap is non-trivial, as directly fine-tuning these models to emphasize such instructions can weaken their overall language capability (Luo et al., 2023). *(ii)* Adapting instruction alignment to different modalities further adds complexity. Each modality imposes distinct demands. For example, visual captions often emphasize object appearance, whereas audio captions may reflect prosodic cues (Sharma et al., 2023; Mei et al., 2022). Meeting heterogeneous requirements usually forces the use of separate models. *(iii)* High-quality training data and benchmarks for omni-modal captioning are also lacking. Most caption datasets cover a single modality (Wang et al., 2024c; Qian et al., 2025). They also provide only coarse-grained captions, which cannot support the fine-grained instruction following required above. In particular, audio data is especially scarce, with existing corpora offering only short COCO-style captions and few long-form descriptions (Kim et al., 2019).

To tackle these challenges, we propose AnyCap, a new captioning framework characterized by two key features. It *(i)* generates captions that are directly controlled by user language instructions, and *(ii)* a single model can handle images, videos, and audio data. Our core idea, inspired by residual learning mechanism in Aligner (Ji et al., 2024), is to refine coarse captions into user-desired ones using a small model. Specifically, we leverage open-source models and available APIs to provide basic captions, and then learn how to align them with user instructions using both the modality observations and instruction texts (Fig. 1). This design simplifies the task, where the base caption supplies general content, while the learning process can focus on capturing instruction intent and refining correctness. Once trained, AnyCap can be used as a plug-and-play module to stack upon diverse captioning models, enhancing captioning without modifying their parameters, or as a flexible tool for re-annotating datasets. It is worth noting that, unlike Aligner that only corrects text modality, AnyCap significantly differs by incorporating multimodal information in a unified framework. Moreover, our aim is to achieve instruction-aligned captioning in an omni-modal setting, which is the first of this kind to our knowledge.

For the lack of instruction-based caption data, we construct a large-scale dataset termed Any-CapData. Unlike prior datasets where instruction types are fixed (*e.g.*, "describe ... in short/details" (Onoe et al., 2024)), we carefully design 28 diverse instruction dimensions spanning images, videos, and audio to better reflect real-world needs (Table 1). To support our alignment paradigm, each sample is annotated as a triplet: an instruction, a preferred caption, and a less-preferred caption, clearly capturing differences in instruction alignment quality. These triplets are generated in a data-aware manner by prompting large MLLMs with the original modality inputs, ensuring that user instructions are grounded in the source content. To further secure quality, we perform small-scale manual screening for each instruction pattern and modality before large-scale generation. With this limited human effort, we improve the quality of automated data generation.

For reliable evaluation, existing methods also remain inadequate. They typically adopt traditional machine translation scores (*e.g.*, BLEU (Papineni et al., 2002), CIDEr (Vedantam et al., 2015)) that lack semantic awareness, or rely on direct LLM-based scoring, which suffers from high randomness across instruction types (Qian et al., 2025; Chen et al., 2025b). We thus design AnyCapEval, with the key insight to categorize diverse instruction types into two orthogonal dimensions, *i.e.*, content and style, and evaluate them separately for focused and fine-grained evaluation. For content evaluation, we introduce the Keypoint Density (KPD) metric. It employs an automatic matcher to measure the recall of content keypoints and penalizes redundancy in excessively long captions. This encourages captions that are both precise and of appropriate length. For style evaluation, we rigorously design detailed scoring rubrics for each instruction type, combined with references to guide evaluation and reduce scoring variance. This structured design enables consistent distinction between compliant and non-compliant stylistic outputs.

Extensive experiments show that AnyCap significantly improves caption quality and instruction-following ability across image, video, and audio. It benefits both open-source and proprietary API-based models. Specifically, stacking a 2B AnyCap on GPT-4o improves content following by 35% and style controllability by 9%. The improvements are also consistently shown on widely-used benchmarks including MIA-Bench and VidCapBench, demonstrating its generalizability. We further validate AnyCap through detailed ablation studies and user studies, observing that, compared to other models, AnyCap tends to convey the instruction-required information more accurately while removing redundant content (Table 11), thereby better matching user preferences.

## 2 RELATED WORK

**Image, video, and audio captioning.** Most early captioning models focus on a single-modality and produce only brief sentences (Vinyals et al., 2015; Venugopalan et al., 2015; Kim et al., 2019). Later research on image and video captioning moves toward longer and finer-grained descriptions (Bianco et al., 2023; Chen et al., 2024a; Ge et al., 2024; Krishna et al., 2017b). Recently, some base MLLMs have attempted to unify image and video captioning (Chen et al., 2024c; Bai et al., 2025) and even cover audio (Hurst et al., 2024; Anil et al., 2023; Xu et al., 2025). Yet, under open-ended prompts these models often drift from user-specific needs, producing irrelevant details or hallucinations that misalign with the intended caption instructions (Han et al., 2024; Huang et al., 2023). To address this, we introduce a plug-and-play model that aligns coarse captions across image, video, and audio into fine-grained, high-utility outputs.

**Fine-grained captioning with instructions.** A range of works use control signals as instructions to specify caption content or style. Representative approaches include employing length embedding for controlling caption length (Zeng et al., 2023), bounding boxes (Cornia et al., 2019) or masks (Wang et al., 2023) for region-level grounding, soft prompting for style or domain transfer (Zhao et al., 2024), and structured prompts to standardize instance-level descriptions (Fan et al., 2024). While effective, these control interfaces are individually designed and difficult to compose, limiting their ability to capture diverse fine-grained requirements. In contrast, we focus on flexible natural language as a unified instruction interface. Our framework aligns captions with 28 instruction types across modalities to broadly cover the spectrum of captioning needs.

**Caption evaluation metrics.** Traditional word-overlap metrics like BLEU (Papineni et al., 2002) and CIDEr (Vedantam et al., 2015) are widely used, but they cannot evaluate semantic consistency or instruction-following ability. This limitation arises because n-gram similarity only measures surface word overlap. Another common approach is to use LLM or MLLM as judges (Maeda et al., 2024; Qian et al., 2025; Chan et al., 2023; Sun et al., 2024; Mao et al., 2024; Dong et al., 2024; Lu et al., 2024; Liu et al., 2023; Sun et al., 2023), which offers semantic coverage but suffers from large variance and model bias (He et al., 2024; Qiu et al., 2023). Recent attempts also tailor metrics toward specificity in long captions (Chen et al., 2025a). We incorporate the strengths of these directions by introducing AnyCapEval benchmark. It explicitly separates *content* and *style* assessment with a particular focus on instruction-following evaluation. We design a new keypoint density (KPD) metric and the constructed scoring rubrics for more reliable assessment of caption quality.

## 3 ANYCAP

### 3.1 FRAMEWORK

Given a modality input $m$ from images ($m^{\text{img}}$), videos ($m^{\text{vid}}$), or audio ($m^{\text{aud}}$), captioning across different modalities often relies on separate MLLMs, and the resulting captions often fail to follow user-specified content or style requirements. To address this, we design AnyCap as a plug-and-play approach to help existing foundation models align captions with user instructions. As illustrated in Fig. 2(a), a frozen base model first processes $m$ to produce an initial caption $y_0$. Then, AnyCap ($\mathcal{M}_a$) takes the original input $m$, the initial caption $y_0$, and a detailed user instruction $q$ to generate a refined caption $y_c$, formulated as

$$y_c = \mathcal{M}_a(m, q, y_0). \tag{1}$$

Concretely, as shown in Fig. 2(b), AnyCap first employs modality-specific encoders (*e.g.*, Intern-ViT (Chen et al., 2024c) for $m^{\text{img}}$ and $m^{\text{vid}}$, and EAT (Chen et al., 2024b) for $m^{\text{aud}}$) to extract

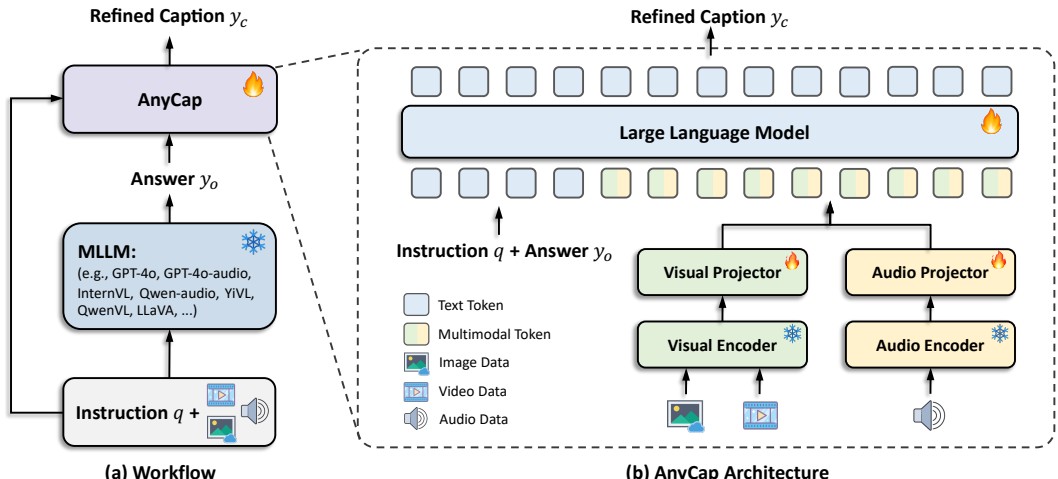

Figure 2: **AnyCap framework.** (a) AnyCap is stacked upon various MLLMs, refining their initial captions into high-quality, instruction-aligned outputs. (b) Specifically, AnyCap takes as input the initial caption, the original modality data, and the user instruction to produce the final caption.

features from each modality input. These modality features are projected into a shared semantic space via modality-specific linear transformations (MLPs). Concurrently, $q$ and $y_0$ are tokenized and embedded into textual embeddings. Finally, modality and textual embeddings are concatenated and fed into a large language model to produce the refined, instruction-compliant caption $y_c$.

The architecture of AnyCap allows us to adopt a residual-correction strategy (Ji et al., 2024). Although supervision still follows standard next-token prediction on the final caption $y_c$, AnyCap focuses on learning the correction pattern from the initial caption $y_0$ to the instruction-compliant target. This refinement is also conditioned on both the modality features and the user instruction, ensuring that the final caption remains grounded in the source input while aligning with the specified requirements. With $y_0$ as reference, we observe that residual learning makes the task easier and delivers two gains: *(i)* it outperforms directly fine-tuning base MLLMs with instructions alone, as shown in Tables 8 and 9; and *(ii)* it enables a single model to be trained jointly across image, video, and audio data. We only train the lightweight plug-and-play AnyCap, leaving the base MLLMs frozen to preserve its original capabilities. To help the model recognize when *no* change is needed, about 40% of training instances already satisfy the instruction. It makes training more stable and mitigates over-editing issue, as ablated in Fig. 4.

## 3.2 ANYCAPDATA

We propose AnyCapData that features three aspects: *(i)* it covers large-scale, high-quality caption data across images, videos, and audio; *(ii)* each caption is paired with user instructions to specify requirements; and *(iii)* each sample adopts a triplet structure to support our residual-correction training, formulated as $(q, c, a)$. Here, $q$ denotes the language instruction, $c$ is a high-quality caption adhering to the instruction, while $a$ is a suboptimal caption that may exhibit minor deficiencies in factual accuracy, level of detail, or compliance with the instruction.

**Instruction types.** To ensure both diversity and practicality, we identify instruction types by systematically surveying caption-related literature to surface common control categories and by analyzing typical downstream requirements. The final selections are further conditioned on the distinctive characteristics of each modality, as listed in Table 1. More details are illustrated in Sec. A.2.

**Construction pipeline.** With the designed instruction types, we first use MLLMs to jointly generate a specific user instruction $q$ and its compliant caption $c$ for each image, video, or audio sample. This avoids cases where a separately predicted instruction might request information the caption cannot provide. The prompts for data construction are tailored per modality and instruction type, including clear task descriptions, explicit length and style constraints, common pitfalls, few-shot exemplars, and the original media as input. For audio, we include the dataset's reference caption (when available) in the prompt to reduce hallucination. Importantly, $q$ is instantiated for each ex-

Table 1: Supported instruction types across image, video, and audio modalities in AnyCapData. Checkmarks (✓) indicate where each control is applicable. *Content* types control what information is conveyed, while *Style* types shape how the message is delivered.

| Instruction Type | Description | Image | Video | Audio |
|---|---|:---:|:---:|:---:|
| *Content* | | | | |
| Background (Bkg) | Provide or suppress scene–background details | | ✓ | |
| Event (Evt) | Require mention of an event or temporal change | | ✓ | ✓ |
| Instance (Ins) | Emphasise or ignore specific entities; enable inter-entity comparison | ✓ | ✓ | |
| Instance Action (IAct) | Describe the motion state of a designated instance | | ✓ | |
| Instance Appearance (IApp) | Characterise visual attributes of a designated instance | ✓ | ✓ | |
| Instance Position (IPos) | Specify spatial position of an instance within the scene | ✓ | ✓ | |
| Movement (Mov) | Specify camera motion type (e.g., pan, zoom, dolly) | | ✓ | |
| Perspective (Per) | Require a particular viewpoint of an object/person | ✓ | ✓ | |
| Region (Reg) | Restrict description to a specified image/video region | ✓ | ✓ | |
| *Style* | | | | |
| Brief (Brf) | Produce a concise rendition with minimal elaboration | ✓ | ✓ | ✓ |
| Detail (Det) | Regulate the required level of descriptive granularity | ✓ | ✓ | |
| Genre (Gen) | Adopt a literary form (e.g., poem (Poe), narrative (Nar)) | ✓ | ✓ | ✓ |
| Length (Len) | Constrain caption size (words or sentences) | ✓ | ✓ | ✓ |
| Theme (Thm) | Conform to a designated linguistic style | ✓ | ✓ | |

ample (*e.g.*, "describe the watermelon's appearance in this image", rather than a generic "describe details"), enabling fine-grained and targeted control. We primarily adopt strong open-source VLMs (*e.g.*, InternVL2.5-78B, Qwen2.5-VL-72B) for visual modality. GPT-4o are selected for harder controls (*e.g.*, perspective, poetry) and for audio modality, balancing coverage and cost. For contrastive supervision, given $(q, c)$ we derive a suboptimal caption $a$ by removing guidance to make it uncontrolled or injecting degradations to increase hallucination. The latter includes omitting required keypoints, introducing mild factual errors, or violating style/length constraints.

Before large-scale automatic construction, each prompt template is manually validated on about 20 instances per modality–type. The validation requires full instruction compliance, no hallucination, correct formatting, and preservation of modality-specific details. After large-scale generation, a check of 5% of the data shows over 95% agreement with human preferences. We also apply type-specific length gate and automatic sanity check to filter degenerate or repetitive outputs. The entire pipeline requires only limited human intervention to enable low-cost scaling.

**Statistics.** The resulting AnyCapData contains about 300k $(q, c, a)$ triplets, spanning images (125k), videos (100k), and audio (75k), constructed from and augmented over roughly 75k samples, which are derived and augmented from 75k original samples. These original samples are collected from multiple public datasets (Wang et al., 2024b; Onoe et al., 2024; Urbanek et al., 2024; Cui et al., 2024; Fan et al., 2024; Kim et al., 2019) with the ensured quality. Data sources and splits, detailed construction procedures, and statistic details are shown in Sec. A.

### 3.3 ANYCAPEVAL

We design AnyCapEval for reliable evaluation of caption quality and, more importantly, the degree of instruction alignment. Beyond previous works, our key idea is that diverse instructions can be grouped into two dimensions: content or style requirements. Accordingly, the evaluation targets: *(i) Content*, which checks whether captions follow the requested control and remain semantically relevant; and *(ii) Style*, which measures consistency with human references in narrative manner, structure, and fluency. They are separately evaluated to prevent mutual conflicts, for example when polished language conceals factual mistakes.

**Content evaluation.** Given a ground-truth caption $c$, a predicted caption $y_c$, and a user instruction $q$, we first annotate a keypoint set $\mathcal{K}_{r,q} = \{k_1, k_2, \ldots, k_n\}$ that exhaustively covers the information required by $q$. GPT-4o then serves as an automatic matcher to identify which subset of keypoints is present in $y_c$ (see Fig. 3(a) for examples). Different from naive approaches that input $(y_c, c, q)$ into an LLM to first find keypoints and then produce a score, we constrain the LLM to a binary classification task, *i.e.*, whether $y_c$ expresses each $k_j$. It simplifies evaluation and improves reliability.

To process verbosity issue, where longer captions may trivially cover more keypoints, we introduce the keypoint density (KPD) to normalize for length, formulated as $\text{KPD}(y_c; q) = N_{\text{match}} / L_{\text{words}}(y_c) \times 100$. $N_{\text{match}}$ is the number of matched keypoints and $L_{\text{words}}(y_c)$ is the word

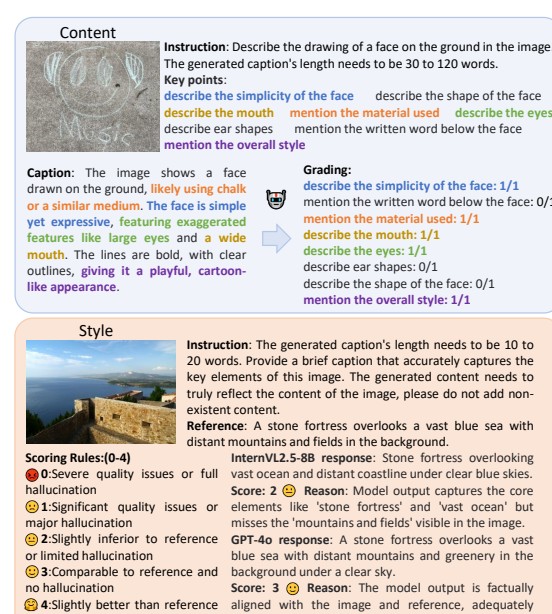

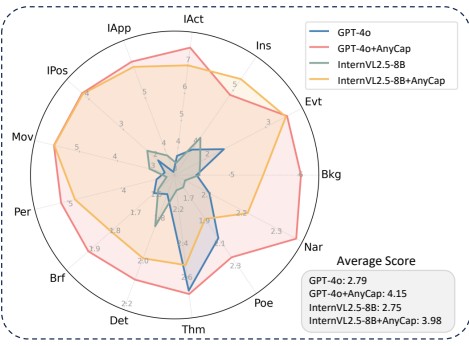

(a) Content & Style Evaluation Examples

(b) Human Validation of AnyCapEval

(c) AnyCap Integration Boosts Base Models' Performance

Figure 3: **Details of AnyCapEval.** (a) Examples for content evaluation via KPD and style evaluation via scoring rules. (b) AnyCapEval judgments highly align with human preference. (c) Integrating AnyCap consistently boosts base models across both content and style dimensions.

count of $y_c$. KPD quantifies the effective information rate, rewarding captions that satisfy user instructions while penalizing redundant or irrelevant content.

**Style evaluation.** We employ GPT-4o to compare the predicted caption $y_c$ with the ground truth $c$ under instruction $q$, producing a discrete score $s(y_c) \in \{0, \ldots, 4\}$. The scoring criteria is as follows: (0) severely deviates from $q$ or largely hallucinated/incorrect; (1) deviates from $q$ or contains many hallucinations; (2) slightly deviates from $q$ or contains certain hallucination; (3) highly similar to $c$ and hallucination-free; and (4) outperforms $c$ while remaining hallucination-free; (see Fig. 3(a)).

This scoring follows explicit criteria along three aspects: semantic similarity, hallucination severity, and stylistic conformity, for each level from 0 to 4. With such a constrained rubric, carefully designed prompts, and side-by-side comparison, the evaluation reduces the judge's degrees of freedom, thereby lowering variance and improving reproducibility.

To verify the bias of AnyCapEval, we conduct a human validation. After assessing multiple models with AnyCapEval, we randomly sample 200 examples per model and ask five independent annotators to judge if the predicted scores and grading rationals (*e.g.*, Fig. 3(a)) aligned with human preference. The results in Fig. 3(b) show high agreement. We also observe that integrating AnyCap consistently improved strong base models in both content and style dimensions (Fig. 3(c)).

## 4 EXPERIMENTS

We train AnyCap in two variants (2B and 8B) on our proposed AnyCapData. Because AnyCap is a relatively small plug-and-play model, it only requires 6 hours on 32 NVIDIA A100 (80GB) GPUs for the 2B variant and 21 hours for the 8B variant to converge. Implementation and training details are provided in Sec. C.1; with a hyperparameter summary in Table 19.

### 4.1 EVALUATION ON INSTRUCTION-ALIGNED CAPTIONING

We evaluate the impact of integrating AnyCap with various powerful base models (proprietary models, *e.g.*, GPT-4o (Hurst et al., 2024); open-source models, *e.g.*, InternVL2.5 (Chen et al., 2024c), Qwen2.5-VL (Bai et al., 2025), MiniCPM-o (Yao et al., 2024)) using our AnyCapEval benchmark across the image, video, and audio modalities. As shown in Tables 2 to 4, adding AnyCap results in

Table 2: Instruction-based image captioning on AnyCapEval. Abbreviations like IPos. are varying instruction types in Table 1. Content and style values are computed using KPD and style scores from Sec. 3.3, respectively. AnyCap significantly improves content accuracy and stylistic fidelity.

| Model | Content ↑ | | | | | Style ↑ | | | | | |
|---|---|---|---|---|---|---|---|---|---|---|---|
| | IPos. | IApp. | Ins. | Per. | **Avg.** | Brf. | Det. | Thm. | Poe. | Nar. | **Avg.** |
| *Proprietary Models* | | | | | | | | | | | |
| GPT-4o | 1.55 | 3.25 | 3.84 | 2.94 | 2.89 | 1.91 | 1.86 | 2.59 | 2.74 | 2.59 | 2.26 |
| +AnyCap-2B | $3.01_{(+1.46)}$ | $4.32_{(+1.07)}$ | $4.55_{(+0.71)}$ | $4.57_{(+1.62)}$ | $4.11_{(+1.22)}$ | $2.29_{(+0.38)}$ | $1.98_{(+0.12)}$ | $2.38_{(-0.21)}$ | $3.02_{(+0.29)}$ | $2.78_{(+0.20)}$ | $2.46_{(+0.19)}$ |
| +AnyCap-8B | $3.48_{(+1.93)}$ | $4.81_{(+1.56)}$ | $4.89_{(+1.06)}$ | $5.00_{(+2.05)}$ | $4.54_{(+1.65)}$ | $2.38_{(+0.47)}$ | $2.35_{(+0.49)}$ | $2.82_{(+0.24)}$ | $3.10_{(+0.37)}$ | $2.87_{(+0.28)}$ | $2.65_{(+0.39)}$ |
| *Open-source Models* | | | | | | | | | | | |
| InternVL2.5-8B | 1.51 | 3.43 | 4.54 | 2.68 | 3.04 | 2.13 | 1.92 | 1.94 | 2.08 | 2.52 | 2.12 |
| +AnyCap-2B | $3.22_{(+1.71)}$ | $4.49_{(+1.06)}$ | $4.82_{(+0.28)}$ | $4.00_{(+1.32)}$ | $4.13_{(+1.09)}$ | $2.27_{(+0.14)}$ | $1.78_{(-0.14)}$ | $2.38_{(+0.44)}$ | $2.71_{(+0.63)}$ | $2.67_{(+0.15)}$ | $2.36_{(+0.24)}$ |
| +AnyCap-8B | $3.41_{(+1.90)}$ | $4.82_{(+1.39)}$ | $4.91_{(+0.37)}$ | $3.89_{(+1.21)}$ | $4.26_{(+1.22)}$ | $2.33_{(+0.20)}$ | $2.10_{(+0.18)}$ | $2.56_{(+0.62)}$ | $2.59_{(+0.51)}$ | $2.83_{(+0.31)}$ | $2.46_{(+0.34)}$ |
| Qwen2.5VL-7B | 1.51 | 3.52 | 5.18 | 3.03 | 3.31 | 2.00 | 2.02 | 2.50 | 2.14 | 2.63 | 2.22 |
| +AnyCap-2B | $3.38_{(+1.87)}$ | $4.42_{(+0.90)}$ | $4.94_{(-0.24)}$ | $4.68_{(+1.65)}$ | $4.36_{(+1.05)}$ | $2.40_{(+0.40)}$ | $2.04_{(+0.02)}$ | $2.50_{(+0.00)}$ | $2.94_{(+0.80)}$ | $2.72_{(+0.09)}$ | $2.50_{(+0.28)}$ |
| +AnyCap-8B | $3.47_{(+1.96)}$ | $4.45_{(+0.93)}$ | $5.84_{(+0.66)}$ | $4.34_{(+1.32)}$ | $4.53_{(+1.22)}$ | $2.36_{(+0.36)}$ | $2.20_{(+0.18)}$ | $2.82_{(+0.32)}$ | $2.88_{(+0.74)}$ | $2.74_{(+0.11)}$ | $2.56_{(+0.34)}$ |

Table 3: Instruction-based video captioning on AnyCapEval. (More details and model comparisons are in Sec. C.3.) AnyCap substantially enhances content adherence and style fidelity.

| Model | Content ↑ | | | | | | Style ↑ | | | | |
|---|---|---|---|---|---|---|---|---|---|---|---|
| | IPos. | IApp. | IAct. | Mov. | Evt. | **Avg.** | Brf. | Det. | Poe. | Nar. | **Avg.** |
| *Proprietary Models* | | | | | | | | | | | |
| GPT-4o | 2.41 | 4.00 | 3.86 | 2.70 | 3.03 | 3.55 | 1.47 | 1.52 | 2.52 | 2.48 | 2.15 |
| +AnyCap-2B | $3.45_{(+1.04)}$ | $6.15_{(+2.15)}$ | $6.26_{(+2.40)}$ | $6.53_{(+3.83)}$ | $4.60_{(+1.57)}$ | $5.30_{(+1.75)}$ | $1.97_{(+0.50)}$ | $1.78_{(+0.26)}$ | $2.60_{(+0.08)}$ | $2.65_{(+0.17)}$ | $2.30_{(+0.15)}$ |
| +AnyCap-8B | $4.92_{(+2.51)}$ | $6.60_{(+2.60)}$ | $7.68_{(+3.82)}$ | $5.67_{(+2.97)}$ | $4.81_{(+1.78)}$ | $5.74_{(+2.19)}$ | $1.95_{(+0.48)}$ | $1.82_{(+0.30)}$ | $2.50_{(-0.02)}$ | $2.77_{(+0.29)}$ | $2.32_{(+0.17)}$ |
| *Open-source Models* | | | | | | | | | | | |
| InternVL2.5-8B | 3.08 | 4.33 | 3.44 | 3.16 | 2.55 | 3.52 | 1.39 | 1.77 | 1.91 | 2.34 | 1.93 |
| +AnyCap-2B | $4.67_{(+1.59)}$ | $6.32_{(+1.99)}$ | $3.19_{(-0.25)}$ | $6.49_{(+3.33)}$ | $4.50_{(+1.95)}$ | $5.07_{(+1.55)}$ | $1.84_{(+0.45)}$ | $1.80_{(+0.03)}$ | $2.48_{(+0.57)}$ | $2.39_{(+0.05)}$ | $2.19_{(+0.26)}$ |
| +AnyCap-8B | $4.95_{(+1.87)}$ | $6.39_{(+2.06)}$ | $7.03_{(+3.59)}$ | $5.66_{(+2.50)}$ | $4.95_{(+2.40)}$ | $5.73_{(+2.21)}$ | $2.00_{(+0.61)}$ | $1.88_{(+0.11)}$ | $2.20_{(+0.29)}$ | $2.61_{(+0.27)}$ | $2.24_{(+0.31)}$ |
| Qwen2.5VL-7B | 2.88 | 4.20 | 3.29 | 2.24 | 2.93 | 3.54 | 1.63 | 1.65 | 2.40 | 2.29 | 2.10 |
| +AnyCap-2B | $3.39_{(+0.51)}$ | $5.97_{(+1.77)}$ | $6.23_{(+2.94)}$ | $6.30_{(+4.06)}$ | $4.48_{(+1.55)}$ | $5.25_{(+1.71)}$ | $1.66_{(+0.03)}$ | $1.82_{(+0.17)}$ | $2.30_{(-0.10)}$ | $2.55_{(+0.26)}$ | $2.16_{(+0.06)}$ |
| +AnyCap-8B | $4.03_{(+1.15)}$ | $6.15_{(+1.95)}$ | $6.58_{(+3.29)}$ | $6.04_{(+3.80)}$ | $4.82_{(+1.89)}$ | $5.55_{(+2.01)}$ | $1.92_{(+0.29)}$ | $1.98_{(+0.33)}$ | $2.45_{(+0.05)}$ | $2.55_{(+0.26)}$ | $2.31_{(+0.21)}$ |

significant improvements in instruction alignment across all modalities and base models, particularly enhancing content fidelity. Fig. 3(c) visualizes these gains for GPT-4o and InternVL2.5-8B.

**Model-centric analysis.** The results consistently show the effectiveness of AnyCap in improving instruction alignment ability. Both the 2B and 8B variants significantly enhance instruction adherence compared to the unassisted base models across all modalities, with the larger AnyCap-8B generally yields greater improvements. Notably, with AnyCap-8B, certain open-source models achieve instruction alignment scores that are comparable or even exceed strong unassisted proprietary baselines. For example, AnyCap-8B elevates the performance of open-source InternVL2.5 from $\sim 2.1$ to $\sim 2.5$ in the image style category. The enhanced score surpasses the $\sim 2.3$ achieved by GPT-4o.

**Task-centric analysis.** Boosts in content-related instruction alignment typically surpass those in style, likely because content controls provide more explicit learning signals, whereas stylistic constraints are inherently more subjective and harder to enforce. Visual tasks outperform audio counterparts, potentially due to the richer supervision and higher information density in vision-language pretraining. AnyCap exhibits varied effects across settings: in well-studied tasks (*e.g.*, image appearance), it refines already strong captions; in less-explored, complex tasks (*e.g.*, video actions or audio events), it fills in missing content, yielding larger gains.

## 4.2 EVALUATION ON PUBLIC BENCHMARKS

To assess AnyCap's generalizability, we integrate it with diverse backbones and evaluate on public image, video, and audio captioning benchmarks. On MIA-Bench (Qian et al., 2025) (Table 5), AnyCap consistently improves performance across models such as InternVL2.5 and Yi-VL (Young et al., 2024). When scaled to AnyCap-8B, it achieves further gains, and even proprietary models such as GPT-4o benefit, reaching new state-of-the-art results. On VidCapBench (Chen et al., 2025b)

Table 4: Instruction-based audio captioning on AnyCapEval. AnyCap consistently improves both content and style metrics over the corresponding base captioners.

| Model | Content ↑ | Style ↑ | | | Avg. |
|---|---|---|---|---|---|
| | Evt. | Brf. | Nar. | Poe. | |
| *Proprietary Models* | | | | | |
| GPT-4o | 1.59 | 1.42 | 1.24 | 0.88 | 1.18 |
| +AnyCap-2B | 1.79$_{(+0.20)}$ | 1.44$_{(+0.02)}$ | 1.38$_{(+0.14)}$ | 1.00$_{(+0.12)}$ | 1.28$_{(+0.10)}$ |
| +AnyCap-8B | 1.88$_{(+0.29)}$ | 1.42$_{(+0.00)}$ | 1.40$_{(+0.16)}$ | 1.08$_{(+0.20)}$ | 1.30$_{(+0.12)}$ |
| GPT-4o mini | 1.28 | 1.17 | 1.16 | 0.79 | 1.04 |
| +AnyCap-2B | 1.56$_{(+0.28)}$ | 1.21$_{(+0.04)}$ | 1.25$_{(+0.09)}$ | 0.89$_{(+0.10)}$ | 1.11$_{(+0.07)}$ |
| +AnyCap-8B | 1.71$_{(+0.43)}$ | 1.10$_{(-0.07)}$ | 1.18$_{(+0.02)}$ | 0.87$_{(+0.08)}$ | 1.05$_{(+0.01)}$ |
| *Open-source Models* | | | | | |
| MiniCPM-o | 1.42 | 1.02 | 0.68 | 0.43 | 0.71 |
| +AnyCap-2B | 1.55$_{(+0.13)}$ | 1.37$_{(+0.35)}$ | 1.32$_{(+0.64)}$ | 0.92$_{(+0.49)}$ | 1.21$_{(+0.50)}$ |
| +AnyCap-8B | 1.48$_{(+0.06)}$ | 1.23$_{(+0.21)}$ | 1.29$_{(+0.61)}$ | 1.13$_{(+0.70)}$ | 1.22$_{(+0.51)}$ |

Table 6: Performance on AudioCaps and Clotho. Metrics: spice (sp), sentence-bert (sbt).

| Model | AudioCaps ↑ | | Clotho ↑ | |
|---|---|---|---|---|
| | sp. | sbt. | sp. | sbt. |
| GPT-4o | 0.06 | 0.41 | 0.07 | 0.37 |
| +AnyCap-2B | 0.06$_{(+0.00)}$ | 0.45$_{(+0.04)}$ | 0.07$_{(+0.00)}$ | 0.40$_{(+0.03)}$ |
| +AnyCap-8B | 0.07$_{(+0.01)}$ | 0.45$_{(+0.04)}$ | 0.08$_{(+0.01)}$ | 0.40$_{(+0.03)}$ |

Table 5: Performance on MIA-Bench. Metrics include accuracy (desc), mention (ment), length (len), perspective (persp), genre (gen).

| Model | desc. ↑ | ment. ↑ | len. ↑ | persp. ↑ | gen. ↑ | avg. ↑ |
|---|---|---|---|---|---|---|
| GPT-4o | 90.3 | 87.9 | 90.5 | 85.2 | 91.6 | 89.1 |
| +AnyCap-2B | 89.1$_{(-1.2)}$ | 88.9$_{(+1.0)}$ | 90.3$_{(-0.2)}$ | 90.0$_{(+4.8)}$ | 92.2$_{(+0.6)}$ | 90.1$_{(+1.0)}$ |
| +AnyCap-8B | 90.4$_{(+0.1)}$ | 89.4$_{(+1.5)}$ | 89.9$_{(-0.6)}$ | 89.2$_{(+4.0)}$ | 92.9$_{(+1.3)}$ | 90.3$_{(+1.2)}$ |
| InternVL2.5-8B | 85.2 | 79.0 | 82.6 | 72.5 | 85.2 | 80.9 |
| +AnyCap-2B | 85.7$_{(+0.5)}$ | 79.6$_{(+0.6)}$ | 85.2$_{(+2.6)}$ | 77.5$_{(+5.0)}$ | 85.7$_{(+0.5)}$ | 82.7$_{(+1.8)}$ |
| +AnyCap-8B | 86.9$_{(+1.7)}$ | 79.0$_{(+0.0)}$ | 86.1$_{(+3.5)}$ | 88.9$_{(+16.4)}$ | 86.3$_{(+1.1)}$ | 85.4$_{(+4.5)}$ |
| Yi-VL-34B | 71.7 | 52.0 | 55.8 | 60.0 | 59.8 | 59.9 |
| +AnyCap-2B | 71.8$_{(+0.1)}$ | 54.2$_{(+2.2)}$ | 61.9$_{(+6.1)}$ | 61.7$_{(+1.7)}$ | 61.0$_{(+1.2)}$ | 62.1$_{(+2.2)}$ |
| +AnyCap-8B | 79.5$_{(+7.8)}$ | 66.5$_{(+14.5)}$ | 67.1$_{(+11.3)}$ | 63.0$_{(+3.0)}$ | 75.8$_{(+16.0)}$ | 70.4$_{(+10.5)}$ |

Table 7: Performance on VidCapBench. Metrics include accuracy (acc), precision (pre), coverage (cov), and conciseness (con).

| Model | acc. ↑ | pre. ↑ | cov. ↑ | con. ↑ |
|---|---|---|---|---|
| GPT-4o | 15.6 | 58.9 | 87.8 | 6.3 |
| +AnyCap-2B | 15.4$_{(-0.2)}$ | 59.1$_{(+0.2)}$ | 87.1$_{(-0.7)}$ | 10.3$_{(+4.0)}$ |
| +AnyCap-8B | 15.8$_{(+0.2)}$ | 59.5$_{(+0.6)}$ | 88.0$_{(+0.2)}$ | 9.3$_{(+3.0)}$ |
| InternVL2.5-8B | 12.8 | 51.6 | 84.2 | 7.1 |
| +AnyCap-2B | 13.8$_{(+1.0)}$ | 55.7$_{(+4.1)}$ | 84.7$_{(+0.5)}$ | 10.1$_{(+3.0)}$ |
| +AnyCap-8B | 14.8$_{(+2.0)}$ | 57.1$_{(+5.5)}$ | 86.9$_{(+2.7)}$ | 10.2$_{(+3.1)}$ |
| Qwen2VL-7B | 13.1 | 53.5 | 82.9 | 6.6 |
| +AnyCap-2B | 14.6$_{(+1.5)}$ | 55.8$_{(+2.3)}$ | 85.0$_{(+2.1)}$ | 10.6$_{(+4.0)}$ |
| +AnyCap-8B | 15.4$_{(+2.3)}$ | 57.3$_{(+3.8)}$ | 86.7$_{(+3.8)}$ | 10.0$_{(+3.4)}$ |

Table 8: AnyCap vs. SFT, DPO, SC on AnyCapEval. Entries are relative improvements (%) over each model's origin baseline under different paradigms, reported separately for content (Cont) and style (Sty). AnyCap demonstrates clear effectiveness.

| Model (Modality) | SFT | | DPO | | SC | | AnyCap-2B | | AnyCap-8B | |
|---|---|---|---|---|---|---|---|---|---|---|
| | Cont. | Sty. | Cont. | Sty. | Cont. | Sty. | Cont. | Sty. | Cont. | Sty. |
| InternVL2.5-8B(Ima) | +9.2 | +3.5 | +9.5 | +4.4 | +3.3 | +0.5 | +35.9 | +11.3 | **+40.1** | **+16.0** |
| InternVL2.5-8B(Vid) | +2.3 | +1.0 | +2.8 | +1.6 | +0.0 | -1.6 | +44.0 | +13.5 | **+62.8** | **+16.1** |
| Qwen2.5VL-7B(Ima) | +6.7 | +3.8 | +7.5 | +4.1 | +11.5 | +5.0 | +31.7 | +12.6 | **+36.9** | **+15.3** |
| Qwen2.5VL-7B(Vid) | +3.2 | +3.0 | +7.3 | +5.2 | +16.7 | **+11.0** | +26.6 | +2.9 | **+56.8** | +10.0 |

Table 9: AnyCap vs. SFT, DPO, SC on MIA benchmark with InternVL2.5-2B.

| SFT | DPO | SC | AnyCap |
|---|---|---|---|
| 1.2% | +7.0% | +2.8% | **+16.9%** |

Table 10: Data ratio comparison with InternVL2.5-8B.

| 1:1:1 | 1:2:2 | 2:1:2 | 2:2:1 |
|---|---|---|---|
| +17.9% | +17.2% | +16.5% | **+21.1%** |

(Table 7), augmenting InternVL2.5-8B with AnyCap yields improvements in accuracy (+2.0), precision (+5.5), and conciseness (+3.1), indicating fewer hallucinations and better capture of motion and content cues. For audio, although references are shorter and metrics more sensitive on Clotho and AudioCaps (Drossos et al., 2020; Kim et al., 2019) (Table 6), AnyCap still consistently enhances fluency and semantic controllability. Collectively, AnyCap reliably steers strong backbones toward more factually grounded captions across modalities.

## 4.3 COMPONENT-WISE ANALYSIS

**Comparison with SFT, DPO, and Self-Critic approaches.** One motivation for plug-and-play AnyCap is to avoid retraining each base model. We compare it against three options: *(i)* supervised fine-tuning (SFT), where instructions are added to prompts to fine-tune the base model; *(ii)* direct preference optimization (DPO) on base models using preferred–non-preferred pairs from AnyCap-Data; and *(iii)* a self-critic (SC) approach where the backbone revises its own outputs at test time. Across both AnyCapEval (Table 8) and MIA (Table 9), AnyCap consistently achieves stronger performance. This advantage arises as it builds on base captions as references, focusing learning on refining outputs toward user instructions rather than generating fully aligned captions from scratch.

**Ablation on training data ratio.** Our triplet training data consist of three main types: ($q$, uncontrolled $a$, $c$), ($q$, $c$, $c$), and ($q$, hallucinated $a$, $c$). To explore the optimal ratio, we first vary the share of fully correct samples ($q$, $c$, $c$). As shown in Fig. 4, the average accuracy on three modalities

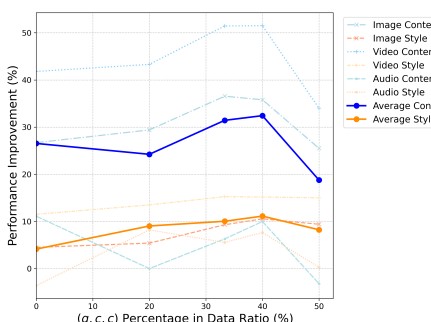

Figure 4: Impact of training data ratio.

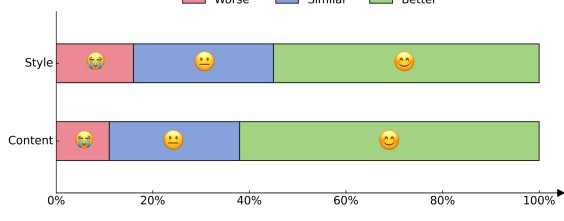

Figure 5: Human evaluation comparing AnyCap-8B with GPT-4o. Captions refined with AnyCap align more closely with the given instructions, enhancing both content and style consistency.

Table 11: Absolute keypoint counts (no length normalization) and response lengths on Any-CapEval image captioning.

| Model | Keypoints (Abs.) ↑ | | | | Response Length | | | |
|---|---|---|---|---|---|---|---|---|
| | Per. | Ins. | IApp. | IPos. | Per. | Ins. | IApp. | IPos. |
| GPT-4o | 2.96 | 3.52 | 3.13 | 1.42 | 95.8 | 91.8 | 92.4 | 93.7 |
| +AnyCap | 2.29 | 4.00 | 3.13 | 1.55 | 49.6 | 83.3 | 61.7 | 51.8 |
| InternVL2.5-8B | 2.54 | 3.64 | 2.85 | 1.21 | 90.0 | 84.6 | 83.0 | 68.0 |
| +AnyCap | 2.44 | 3.76 | 2.85 | 1.70 | 62.5 | 76.6 | 59.4 | 49.8 |
| Qwen2.5VL-7B | 2.68 | 3.80 | 2.82 | 1.21 | 84.0 | 70.3 | 79.3 | 74.2 |
| +AnyCap | 2.59 | 3.78 | 2.95 | 1.64 | 64.0 | 68.0 | 66.7 | 51.6 |

Table 12: Human evaluation on AnyCapEval comparing manually written captions and Any-Cap. AnyCap matches non-experts but remains slightly below expert performance, indicating clear headroom for future improvement.

| | Image ↑ | | Video ↑ | | Audio ↑ | |
|---|---|---|---|---|---|---|
| | Cont. | Sty. | Cont. | Sty. | Cont. | Sty. |
| Expert | 4.62 | 2.77 | 6.04 | 2.62 | 3.30 | 4.13 |
| Non-expert | 4.42 | 2.66 | 5.72 | 2.35 | 2.01 | 1.37 |
| GPT+AnyCap | 4.54 | 2.65 | 5.74 | 2.32 | 1.88 | 1.30 |

improves steadily up to around $40\%$, but declines when exceeding $50\%$. This suggests that although exposure to correct captions benefits learning, excessive reliance may lead to overfitting.

We further perform ablation across all three types. Results in Table 10 show that including more uncontrolled $a$ samples yields better performance than emphasizing hallucinated $a$. A likely reason is that outputs from strong base models are more prone to instruction violations than factual errors, so AnyCap gains more from frequently correcting uncontrolled cases.

**Keypoint coverage v.s. caption length.** In Table 11, we analyze the relationship between caption length and keypoint coverage. An important property of AnyCap is that it does not "win by writing more". Instead, it effectively reduces unrelated details from base models (especially GPT-4o, as visualized in Fig. 6) while maintaining or even improving keypoint accuracy. The ability to generate concise yet accurate captions is one reason why AnyCap achieve higher scores on AnyCapEval.

### 4.4 HUMAN EVALUATION

To complement model-based evaluation, we conduct two-fold human study. First, a preference test (Fig. 5) compare AnyCap-8B with GPT-4o on both visual and auditory modalities, asking annotators which better satisfied content and style criteria. Second, we hire PhD-level humanities experts and well-educated non-experts to handwrite captions for comparison with AnyCap. As shown in Table 12, experts surpassed AnyCap in fine control, while AnyCap matched or slightly outperformed the non-expert group on visual tasks and remained broadly comparable on audio. This suggests that AnyCap delivers strong quality while aligning well with human preferences.

### 5 CONCLUSION

We present AnyCap, a lightweight module that aligns base captioners with human instructions in omni-modal settings, enhancing instruction-following without retraining base models. It leverages residual correction in a unified multimodal design, enabling a single model to serve images, videos, and audio. To support this, we introduce AnyCapData, a large-scale, fine-grained instruction–based caption dataset. We also present AnyCapEval, a new benchmark to more accurately evaluate instruction alignment across modalities. On both public benchmarks and AnyCapEval, AnyCap consistently achieves superior performance, advancing instruction-aligned captioning across modalities.

**Ethics statement.** While AnyCapData is carefully curated to support fine-grained controllability, there remains a risk of malicious misuse (for instance, deliberately creating harmful or misleading Q-A pairs). Models trained on such compromised data could exhibit significantly degraded controllability, resulting in severe negative impacts. We urge caution and responsible practices when utilizing or expanding upon our resources.

**Reproducibility statement.** The framework and training strategy of AnyCap are described in Sec. 3.1, with detailed training settings and hyperparameters provided in Sec. C.1. The construction process, advantage comparison, data statistics, and quality assurance of AnyCapData are outlined in Sec. 3.2, with additional details and the required prompt templates provided in Sec. A. The testing process of AnyCapEval is fully described in Sec. 3.3. More details of AnyCapEval are discussed in Sec. B, including an analysis of its correlation with human judgments. The experimental setup and more detailed results for some experiments are provided in Sec. C, where we also include visualizations of the experimental outcomes and additional performance details of AnyCap on downstream tasks. All code and data will be open-sourced to ensure reproducibility and further research.

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

APPENDIX

# A DATASETS

## A.1 OVERVIEW OF CAPTION DATASETS

Automatic captioning for modalities such as images, videos, and audio is a core task in artificial intelligence, aiming to enable machines to understand and describe perceived content using natural language, much like humans do. The foundation supporting this research is large-scale annotated datasets. However, existing datasets (Table 13) exhibit significant limitations in providing fine-grained, multi-dimensional control capabilities. We next detail the motivation and data analysis behind our dataset.

Table 13: Overview of the publicly available caption datasets analyzed in this study. "Modality" indicates the primary input type (e.g., image, video, or audio), while "Instruction Type" refers to the predominant linguistic or structural annotation style as described by the original dataset authors.

| Dataset | Modality | Instruction Type |
|---|---|---|
| ASD v2 | Image | Relation |
| MDVP-Data | Image | Region / Brief / Detail |
| DCI | Image | Dense |
| DOCCI | Image | Dense |
| ImageInWords (IIW) | Image | Dense |
| ShareGPT-4o | Image | Detail |
| ShareGPT-4v | Image | Detail |
| ShareGPT-4o (video) | Video | Detail |
| MSR-VTT | Video | Brief |
| MSVD (YouTube2Text) | Video | Brief |
| VATEX | Video | Brief |
| Video-ChatGPT (VideoInstruct-100K) | Video | Dense |
| InstanceCap / InstanceVid | Video | Instance-level (structured) |
| ShareGPT4Video | Video | Detail |
| MiraData | Video | Structured |
| LLaVA-Video-178K | Video | Detail |
| AudioCaps | Audio | Brief |
| Clotho | Audio | Brief |
| MACS | Audio | Brief |
| WavCaps | Audio | Brief |

## A.2 THE SINGULARITY AND SCARCITY OF USER INSTRUCTIONS

A "user instruction" is an input accompanying the main media (image, video, or audio), intended to guide or constrain the generation of descriptive content. Ideally, we desire models capable of generating descriptions with varying styles, focusing on different aspects, and meeting specific requirements based on diverse user instructions.

Most classic datasets, such as COCO (Lin et al., 2014) and Flickr30k (Young et al., 2014) in the image domain, MSR-VTT (Xu et al., 2016) and MSVD (Chen & Dolan, 2011) in the video domain, and AudioCaps (Kim et al., 2019) and Clotho (Drossos et al., 2020) in the audio domain, primarily provide "media–caption pairs". They typically include multiple reference captions provided by different annotators. This "multi-reference" design is mainly intended to evaluate the diversity and coverage of descriptions, rather than providing explicit control during generation. The model's primary task is to generate a "reasonable" description, but the user cannot specify requirements at generation time (*e.g.*, "generate a humorous caption", "describe the background elements in detail"). The only implicit user instruction is essentially "describe this content in brief".

Some datasets offer a degree of control, but often only along a single dimension. Dense captioning datasets, like DCI (Urbanek et al., 2024) for images or video datasets providing temporal informa-

tion (Krishna et al., 2017a), associate descriptions with specific image regions or video segments. The "region/time segment" here can be viewed as a user instruction, *i.e.*, "describe *this part*". However, this is usually the sole dimension of control. Datasets like ASD v2 (Wang et al., 2024b) focus on describing relationships between entities in an image. The user instruction is the "entity pair" whose relationship needs to be described. Datasets guided by concepts like MiraData (Ju et al., 2024) or InstanceCap (Fan et al., 2024) require the output to follow a specific structure, which constitutes a form of control over the output format. Although such structured outputs often encompass several instruction types (*e.g.*, Background, Detail), the formulation and granularity of these user instructions are typically fixed, lacking the flexibility for customized control requests tailored to each individual image, video or audio.

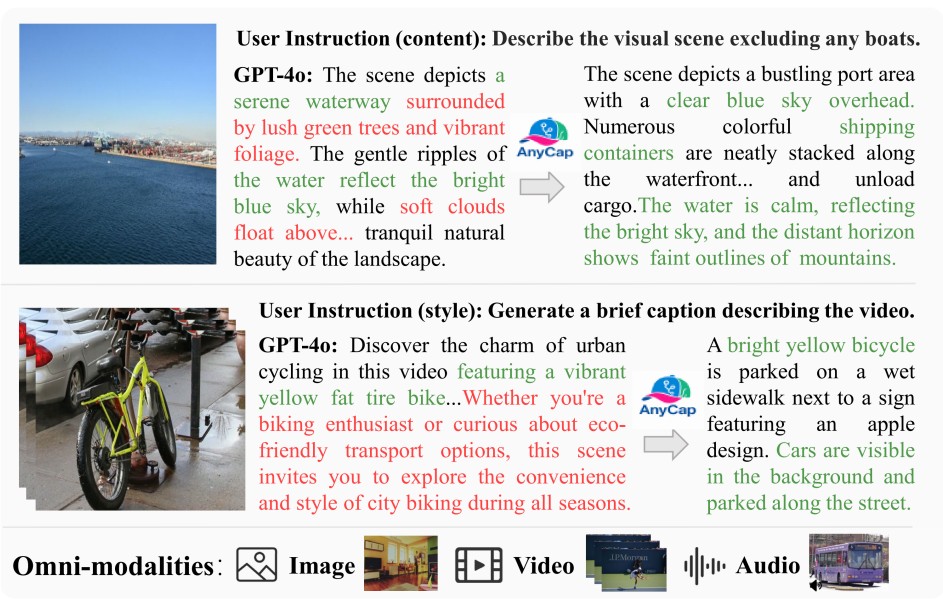

Figure 6: AnyCap enables controllable captioning across modalities by refining base model outputs to better align with user instructions. Given a user instruction, it takes initial captions from a foundation model and corrects instruction violations (highlighted in red), producing compliant, instruction-following outputs (green), all without requiring fine-tuning of the base model.

In summary, the majority of existing public datasets lack the ability to align captions with specific user instructions across multiple dimensions, limiting the development and evaluation of instruction-aligned captioning technologies. Even when leveraging large language models such as GPT for annotation, the resulting captions often reflect general patterns and fail to align precisely with the specified instructions along desired dimensions.

To address this gap, we construct AnyCapData, a large-scale dataset featuring diverse user instructions paired with high-quality, instruction-compliant captions across multiple modalities. Leveraging this dataset, we train AnyCapModel to achieve fine-grained instruction-aligned caption generation. As illustrated in Fig. 6, AnyCap can transform initial captions into highly specific descriptions that accurately follow various user instructions, enabling precise and customizable outputs without requiring modifications to the base model architecture.

### A.3 DATA ANALYSIS

**Data sources.** To construct AnyCapData, we utilize multiple publicly available datasets spanning three modalities: images, videos, and audio. Specifically, we incorporate data from ASD v2, DCI, DOCCI, ShareGPT-4o, InstanceCap, and AudioCaps. From these datasets, we extract images, videos, and audio content, and employ various multimodal large language models (MLLMs) to generate diverse user instructions and captions based on predefined instructions.

**Data statistics.** AnyCapData encompasses three primary modalities: images, videos, and audio, comprising a total of 300k $(q, c, a)$ triplets. Among them, 125k belong to the image modality, 100k to video modality, and 75k to audio modality. Each modality includes multiple instruction types.

Table 14: Instruction prompt template for generating user instructions and high-quality video captions. This prompt is designed to elicit high-quality instruction–caption pairs $(q, c)$ from video content, where $c$ serves as an optimally grounded caption based strictly on observable visual evidence. The template enforces modality-specific constraints to ensure factual and non-speculative outputs for optimal generation results.

---

**Video Understanding Expert Instructions**
You are an AI expert in video content analysis, specializing in generating precise and structured descriptions based ONLY on directly observable video content. Your core capabilities include analyzing video elements such as actions, objects, scenes, interactions, and camera movements to provide factual, observation-based descriptions.

**Your Task**
Generate concise captions that capture ONLY the directly visible and essential content of videos according to specific constraints. You must analyze only the observable content and ensure all descriptions are based on concrete visual evidence rather than assumptions or inferences.

**Key Guidelines for Caption Generation**
1. The output question formats can be varied
2. Create concise descriptions that capture only directly observable essential elements
3. Focus on analyzing ONLY these visible components:
- Actions and events shown in frame
- Objects and characters physically present
- Scene settings visible in shot
- Camera movements and angles that can be seen
- Interactions occurring on screen
4. DO NOT include:
- Assumptions about off-screen elements
- Inferences about motivations or thoughts
- Speculation about context or background
- Interpretations of meaning
- Details that cannot be directly seen
- Guesses about what happened before/after
5. Maintain strict adherence to:
- Only describing what is visually present
- Excluding all speculative content
- Basing every detail on visual evidence
- Using clear, objective language

**Constraints**
- Descriptions MUST be based EXCLUSIVELY on visible content
- NO assumptions or interpretations beyond what can be directly seen
- EXCLUDE any details that require inference or speculation
- Keep descriptions concise and focused on key visible elements
- Follow any additional specific requirements provided with each request

**Examples:**
Input 1:
Question: Generate a brief caption describing this video's main content.
Answer: Two news anchors engage in an animated discussion at a professional news desk, maintaining eye contact while exchanging viewpoints.
Input 2:
Question: Use a brief caption to convey the main scene or content of the video.
Answer: A person wearing a blue shirt walks leisurely along a scenic path bordered by tall, leafy trees.

**Output Format**
Question: [Your various question, but need to express the generation of a brief caption]
Answer: [Your concise description based STRICTLY on visible content with NO speculation]

---

**Generation of instruction-caption triplets.** To generate the $(q, c, a)$ triplets in AnyCapData, where $q$ denotes a user instruction, $c$ is a high-quality caption, and $a$ is a relatively suboptimal caption, we utilize several MLLMs guided by specifically designed instruction templates. The de-

Table 15: Instruction prompt template for generating suboptimal captions $a$ from high-quality instruction–caption pairs. For a given instruction $q$ and its high-quality caption $c$, the template produces slightly degraded variants by introducing controlled inaccuracies (e.g., minor omissions or speculative insertions) while maintaining semantic coherence with the original content. Such prompts facilitate contrastive training and robustness evaluation in multimodal scenarios.

---

**Video Understanding Expert Instructions**

You are an AI expert in video content analysis, specializing in generating precise and structured descriptions based on specific queries. However, your task now includes generating slightly inferior captions that introduce minor inaccuracies or deviations while still maintaining general relevance to the video content.

**Your Task**

Generate slightly inferior captions based on given examples:
- Adding minor inaccuracies.
- Omitting a small but relevant detail.
- Including an unnecessary or speculative element.
- Misinterpreting a minor aspect of the video.
- Does not meet the requirements of the question.

**Key Guidelines for Caption Generation**

1. Base the description on the standard caption but allow slight deviations:
- Slightly altering actions, movements, or interactions in the video
- Adding irrelevant or speculative elements about the scene or context
- Omitting small but observable details from the footage
- Misinterpreting temporal sequences or duration
2. Ensure the captions remain generally related but slightly inferior to the standard caption
3. Focus on introducing appropriate levels of inaccuracy while maintaining plausibility

**Constraints**

- Deviations must be minor and not completely distort the video's main content.
- Avoid making the caption entirely incorrect or irrelevant.
- Ensure the captions remain plausible and connected to the visible elements in the video.
- Consider the temporal nature of video content when introducing inaccuracies.

**Examples:**

Input 1:
Question: Generate a brief caption describing this video's main content.
Standard Caption: Two news anchors engage in an animated discussion at a professional news desk, maintaining eye contact while exchanging viewpoints.
Generated Caption: Three news anchors engage in a discussion at a news desk, but the situation appears to be getting out of control with potential physical confrontation.
Input 2:
Question: Use a brief caption to convey the main scene or content of the video.
Standard Caption: A person wearing a blue shirt walks leisurely along a scenic path bordered by tall, leafy trees.
Generated Caption: A young man in a green long-sleeve shirt runs along a scenic path lined with dense trees.

**Output Format**

You MUST strictly adhere to this format:
Question: {question}
Standard Caption: {standard_caption}
Generated Caption: [Your generated caption with appropriate deviations]

---

tailed templates employed in this work are presented in Table 14 and Table 15. These templates are constructed according to different instruction types to ensure structured generation and facilitate downstream analysis. Each raw data sample is paired with one or more tailored instructions to produce diverse outputs.

## A.4   FUTURE DIRECTIONS

Future dataset development could focus on the following improvements:

**Constructing datasets with rich user instructions:** There is a need to design new datasets containing diverse, explicit, and composable instruction types.

**Leveraging synthetic data:** Explore the use of large language models to generate synthetic description data with specific user instructions, serving as a supplement to real data.

**Improving evaluation methods:** Develop new evaluation protocols and metrics capable of assessing fine-grained aspects like controllability, style consistency, and faithfulness.

# B ANYCAPEVAL BENCHMARK

## B.1 LIMITATIONS OF EXISTING EVALUATION METHODS

**Limitations of traditional machine translation metrics.** Metrics developed for machine translation, such as BLEU (Papineni et al., 2002), ROUGE (Lin, 2004), METEOR (Banerjee & Lavie, 2005), and CIDEr (Vedantam et al., 2015), primarily focus on n-gram overlap or co-occurrence statistics. While they can measure fluency and lexical similarity, they fail to accurately capture semantic consistency. For instance, sentences like "I do love large models" and "I do not love large models" could receive similarly high scores, despite conveying opposite meanings. These metrics cannot effectively detect deep semantic divergence, factual errors, or noncompliance with user instructions.

**Challenges of multimodal large language models scoring.** Recent evaluation schemes leverage powerful language models or multimodal large language models (MLLMs) to directly score captions. However, these methods suffer from significant randomness and instability: the same caption might receive drastically different scores across minor prompt variations or multiple queries. Additionally, hallucinated or semantically incorrect captions may still receive high scores, undermining the reliability of such automatic evaluation.

**Instability of keypoint extraction-based metrics.** Some studies, including those utilizing benchmarks like (Wang et al., 2024a) which aims for fine-grained descriptions, propose extracting key information points from captions using MLLMs and computing precision, recall, and F1 scores. However, keypoint extraction itself is unstable: the number and granularity of extracted points vary widely across runs. Our empirical analysis shows low correlation between model-extracted keypoints and human-annotated ones, suggesting that such extraction-based metrics poorly reflect the true information content of captions.

**Challenges of QA-pair based evaluation.** Benchmarks such as VDC (Chai et al., 2024) highlight that QA-pair based evaluation suffers from instability, primarily due to the variable quality and granularity of the QA pairs themselves. Designing high-quality, representative QA items that faithfully capture fine-grained caption attributes remains an open challenge.

## B.2 RATIONALE BEHIND ANYCAPEVAL DESIGN

### B.2.1 SEPARATION OF CONTENT AND STYLE

Through extensive analysis of user instruction types in captioning, we divide evaluation into two orthogonal dimensions:

- **Content:** We expect the generated captions to strictly adhere to the given user instructions, maintain clear topical relevance, and avoid introducing irrelevant information. To evaluate this aspect, we employ keypoint density (KPD) as the primary metric.
- **Style:** We aim for the generated captions to closely match human-crafted references in terms of narrative style, tone, and expressive form. To assess this, we design a fine-grained scoring system.

This separation is crucial because content and style represent fundamentally different aspects of caption quality that require distinct evaluation approaches. Content adherence focuses on factual accuracy and instruction compliance: whether the caption correctly follows the user's specified requirements about what to describe. This is objectively verifiable through user instruction alignment. In contrast, style evaluation assesses subjective qualities like fluency, coherence, and naturalness that

reflect human-like expression. Merging these dimensions would conflate objective and subjective criteria, making it difficult to diagnose specific model weaknesses. The orthogonal evaluation allows for more precise identification of whether a model's limitations lie in its ability to follow instructions (content) or in its linguistic quality (style), enabling targeted improvements to each aspect. Furthermore, different applications may prioritize these dimensions differently: some use cases demand strict content control while others emphasize stylistic quality, making separate evaluation essential for practical deployment decisions.

### B.2.2 KEYPOINT DENSITY METRIC

**Intuitive motivation.** We expect the generated captions to efficiently embed as many control-required key information points as possible within a limited textual length, while minimizing irrelevant or redundant content. This analogy is comparable to adding an appropriate amount of solute into a limited solvent to achieve a targeted concentration. Excessively diluting it in a large solvent volume, even if the absolute quantity increases, results in negligible concentration and ineffective outcomes. To prevent models from exploiting this metric by generating excessively short captions that artificially inflate density scores, we normalize the number of extracted keypoints by the caption length (or an appropriate normalization factor) and apply penalties to captions whose lengths fall outside the predefined acceptable range.

**Length Normalization.** We randomly sample approximately 200 caption instances and extract information points using both GPT-4o (including both control-required keypoints and irrelevant information points) and human annotators. The human-annotated information points are obtained by post-processing the model-generated information points through manual filtering and revision. Correlation analyses, presented in the main paper, reveal that caption length exhibits a significantly stronger association with human-annotated information content than the number of keypoints automatically extracted by the model. This empirical evidence further supports the rationality of applying length normalization in the evaluation metric.

| Correlation Type | Info Points vs. Actual Info | Caption Length vs. Actual Info |
|---|---|---|
| Pearson | 0.284 | 0.521 |
| Spearman | 0.256 | 0.461 |
| Kendall's Tau | 0.231 | 0.344 |

Table 16: Correlations Comparison

### B.2.3 GPT-4O AS THE EVALUATOR

When using GPT-4o as the evaluator, we carefully design structured prompts that explicitly instruct the model to assess Content and Style separately, ensuring consistent and interpretable judgments. See Table 17 and Table 18 for the detailed prompts used for Content and Style evaluation.

## C ADDITIONAL EXPERIMENTAL DETAILS

### C.1 EXPERIMENTAL SETUP

The experimental setup utilizes a computing infrastructure comprising 32 NVIDIA A100 GPUs, distributed across 4 nodes with 8 GPUs each. Each GPU is allocated 10 CPU cores, and all training jobs are managed using Slurm's `srun` with GPU reservation. For distributed training, we employ `torch.distributed`, launching via `srun` to enable dynamic rank assignment and multi-node coordination. We train AnyCapModel on AnyCapData with the AdamW optimizer for 3 epochs, a learning rate of $1 \times 10^{-6}$, a cosine learning-rate schedule with a 0.03 warmup ratio, weight decay of 0.01, a global batch size of 256, and bfloat16 mixed precision. As the modality backbones we use InternVL (vision) and EAT (audio). InternVL's open training and fine-tuning pipeline facilitates hyperparameter tuning, and EAT attains state-of-the-art results on public benchmarks, making both solid foundations for our experiments. During AnyCap training, the external modality backbones are frozen, while the AnyCap internal components, including its language model and MLP layers, are

Table 17: Evaluation instruction for content-controlled video captioning in AnyCapEval.

---

**Task**

You are a video-caption evaluation expert. You will be provided with a video, a caption describing the video, and a set of key points outlining important aspects of the video. Your task is to evaluate whether the caption mentions and accurately describes the given key points based on the video.

**Task Steps**

For each key point, follow these steps:
Step 1: Check whether the key point is mentioned in the caption.
- If the key point is not mentioned, assign a score of 0. - If the key point is mentioned (either exactly or with semantically similar phrasing), proceed to Step 2.
Step 2: Determine whether the description of the key points is correct.
- If the description aligns with or is semantically equivalent to the key point, assign a score of 1. - If the description is incorrect, or does not accurately fit the key point, assign a score of 0.

**Evaluation Constraints**

1. No Assumptions Beyond the Key Point: Only evaluate what is mentioned in the key point. Do not infer additional details not explicitly depicted.
2. Semantic Similarity Allowed: Phrases with similar meaning should be considered matches (e.g., "holding a ball" and "grasping a sphere").
3. Consistent Evaluation: Apply the same evaluation criteria to all key points to ensure fairness and uniformity.

**Scoring Report**

Return the format with:
{"caption_evaluation": {"key_points_scores": {"key_point_1": score, "key_point_2": score, ...}, "total_score": sum_of_scores, "score_reasons": {"key_point_1": "reason for score", "key_point_2": "reason for score", ...}}}

**Example Input**

Key points:
mention the man's position (standing on the left side of the table)
describe the man's appearance (wearing glasses)
mention the woman's position (sitting on the right side of the table)
describe the woman's appearance (wearing a red dress)
mention the boy's position (crouching under the table)
describe the boy's action (picking up a toy)
Caption: A man stands on the left, a woman sits on the right, and a boy is under the table.

**Example Output**

{"caption_evaluation": {
"key_points_scores": {"mention the man's position": 1, "describe the man's appearance": 0, "mention the woman's position": 1, "describe the woman's appearance": 0, "mention the boy's position": 1, "describe the boy's action": 0},
"total_score": 3,
"score_reasons": {"mention the man's position": "Correctly mentions standing on the left side of the table", "describe the man's appearance": "Missing glasses reference", "mention the woman's position": "Correctly mentions sitting on the right side of the table", "describe the woman's appearance": "Missing red dress reference", "mention the boy's position": "Correctly mentions crouching under the table", "describe the boy's action": "Missing picking up a toy reference"}
}}

**Input**

Key points: {key_points}
Caption: {answer}

**Output**

Please return the output exactly as in the example above, without adding anything else.

---

updated. For images, we enforce an input size of 448 pixels with a maximum dynamic patch count of 6 and a down-sampling ratio of 0.5; dynamic image sizing and thumbnail support are enabled. The detailed hyperparameters used during training are listed in Table 19.

## C.2 ABLATION STUDY

**Impact of training data ratio configurations.** The base model for image and video modalities is InternVL2.5-8B. Since InternVL2.5-8B does not support audio modality, we use GPT-4o as a substitute. We begin by introducing the data ratio notation. For instance, in the configuration 2B-122,

the triplet 122 specifies the composition ratio of three sample types used for a given modality. Each digit indicates the relative proportion of a specific sample type.

The three sample types encoded by each digit (from left to right) are defined as follows:

- The first digit indicates the proportion of samples of type ($q$, uncontrolled $a$, $c$), where the caption $a$ is not fully compliant with the instruction $q$, or where the model is expected to directly generate the target caption $c$.

- The second digit indicates the proportion of samples of type ($q$, $c$, $c$), where the initial caption already matches the desired controlled caption or is factually correct.

- The third digit indicates the proportion of samples of type ($q$, hallucinated $a$, $c$), where caption $a$ contains hallucinated content, meaning it describes information not present in the multimodal input.

Table 20 presents the overall performance comparison. Detailed ablation results for each modality and instruction type are presented in the following tables:

- Image modality: Content and Style control in Table 21.
- Video modality: Content control in Table 22, Style control in Table 23.
- Audio modality: Content and Style control in Table 24.

### C.3 FURTHER RESULTS FOR THE MAIN EXPERIMENTS

In the main paper we reported the performance of representative models on AnyCapEval. Here we provide a broader set of results ( Tables 25 to 27). Note that Tarsier2-7B, while strong at detailed captioning, attains comparatively lower scores on our benchmark because it tends to produce long, highly detailed descriptions that do not strictly follow the given instructions, indicating limited controllability. By contrast, AnyCap makes explicit use of rich instruction signals and achieves substantially better controllability and keypoint coverage. Concretely, with AnyCap we observe point counts of Content: 6.18 and Style: 2.26, compared with GPT-4o's Content: 3.55 and Style: 2.15 under the same protocol.

To complement the main experimental results presented in the paper, we include qualitative examples that demonstrate the effectiveness of our controllable generation approach across images, videos and audio modalities. These examples visually and intuitively show how the model performs under different user instructions, including both **Content** and **Style** conditions, as discussed in the main paper. By providing representative outputs, we aim to offer a clearer sense of the generation behavior and control precision achieved by our method. All examples correspond to the settings evaluated in the main experiments and are included here for qualitative inspection. Representative qualitative results for each modality are shown in Fig. 7, Fig. 8, and Fig. 9.

### C.4 DOWNSTREAM UTILITY

We further evaluate AnyCap's utility in enhancing noisy captions for downstream video and image generation. Applying AnyCap to refine captions, including those from the Panda video dataset (Wang et al., 2020), we observe that the resulting video generations demonstrate improved visual-semantic grounding and alignment compared to those using the original, unrefined captions (Wan et al., 2025; Kong et al., 2024). Quantitative results for video generation are presented in Table 28. These findings underscore the effectiveness of AnyCap in real-world multimodal generation pipelines.

To complement the quantitative results, we provide qualitative examples showcasing the impact of caption refinement by AnyCap on downstream image and video generation tasks. Specifically, we compare generations produced using initial captions with those using captions refined by AnyCap. As shown in Fig. 10 and Fig. 11, refined captions lead to outputs that exhibit more accurate visual-semantic correspondence, richer scene details, and improved coherence. These examples further highlight the practical utility of AnyCap in enhancing multimodal generation quality in real-world applications.

## D    USE OF LARGE LANGUAGE MODELS

In preparing this manuscript, we employed advanced language models (*e.g.*, GPT-5, OpenAI, 2025) solely as editorial aids. Their role was limited to refining phrasing, improving readability, and smoothing stylistic inconsistencies across sections. The models were not engaged in developing research questions, proposing methods, analyzing results, or forming conclusions. All substantive ideas, experimental protocols, and technical contributions originated from the authors. Every sentence revised with model assistance was subsequently inspected and approved by human co-authors.

Table 18: Evaluation instruction for style-controlled video captioning in AnyCapEval.

**Task**
You are an expert evaluator tasked with scoring model outputs based on a specific rubric. Please carefully analyze the provided video frames, "Caption Type", "Model Output," and "Reference" text, and assign a score between 0 and 4 to "Model Output" using different criteria for different caption types. Ensure your scoring is consistent and strictly adheres to the definitions provided.

**Scoring Rubric**
- 0 (Very Poor): Severe quality issues OR full hallucination (100% of the content is irrelevant to the facts).
- 1 (Poor): Significant quality issues OR major hallucination (¿50% of the content is fictitious or contradictory).
- 2 (Below Average): Slightly inferior to reference OR limited hallucination (¡50% of the content is inaccurate, but does not affect the core content).
- 3 (Good): Comparable to reference AND no hallucination (factually aligned).
- 4 (Excellent): Slightly better than reference AND no hallucination (factually flawless).

**Caption Type Definitions and Quality Criteria**
1. brief:
- High Quality: Length is within ±30% of the reference word count; concise and captures the core content of the video.
- Low Quality: Length exceeds ±30% of the reference word count; includes irrelevant details or omits key information.
2. detail:
- High Quality: Length is within ±30% of the reference word count; provides rich descriptions of the video's main elements, actions, and settings.
- Low Quality: Length exceeds ±30% of the reference word count; descriptions lack detail or include irrelevant information.
3. poem:
- High Quality: Format and content align closely with the reference; follows poetic conventions (e.g., rhyme, rhythm, line breaks) and is relevant to the video's theme.
- Low Quality: Format and content differ significantly from the reference; disjointed or lacks poetic quality.
4. narrative:
- High Quality: Format and content align closely with the reference; presents a coherent narrative with elements like time, place, characters, and events shown in the video.
- Low Quality: Format and content differ significantly from the reference; disjointed or lacks key narrative elements.
5. style:
- High Quality: Style and content align closely with the reference; matches the narrative style (e.g., humorous, serious, romantic) and is relevant to the video's theme.
- Low Quality: Style and content differ significantly from the reference; mismatched style or irrelevant to the theme.

**Instructions**
1. Compare the model output with the reference text to determine the quality of the model output.
2. Compare the model output to the video frames to determine the severity of the hallucination.
3. For caption quality, evaluate based on:
- The Quality Criteria mentioned above.
- For Caption Type that is brief or detail, ensure the model output's word count is within ±30% of the reference word count. If not, the score cannot be higher than 1 for brief and detailed captions.
- Alignment: Check alignment with the reference in format, style, and content.
4. For hallucination, evaluate based on:
- Factual accuracy and relevance to the video content.
- Consider temporal aspects and action sequences shown in the video frames.
5. Assign the most appropriate score (0-4) based on the rubric.
Mandatory Rule: For the Caption Type that is brief or detail, if the length exceeds ±30% of the reference word count, the score cannot be higher than 1.
6. Return your response in this format:
{"score": [0-4], "reason": "1-2 sentence explanation"}

**Input**
Caption Type: {caption_type}
Model Output: {output}
Reference: {reference}

**Output**
Please strictly return the output in the above format and do not add any other content.

Table 19: Key hyperparameters used for training AnyCap.

| Parameter | Value |
|---|---|
| Per-device batch size | 1 |
| Gradient accumulation steps | 1 |
| Learning rate | $1 \times 10^{-6}$ |
| Weight decay | 0.01 |
| LR scheduler | Cosine |
| Warmup ratio | 0.03 |
| Training epochs | 3 |
| Max sequence length | 4096 |
| Drop path rate | 0.1 |
| Dataloader workers | 4 |
| Gradient checkpointing | Enabled |
| Mixed precision | BF16 |

Table 20: Ablation results on different training data ratio configurations using AnyCapEval, evaluated across Content and Style on images, videos, and audio modalities. All percentage values represent improvements over the base model (InternVL2.5-8B).

| Data Ratio | Image | | Video | | Audio | | Average | |
|---|---|---|---|---|---|---|---|---|
| | Content | Style | Content | Style | Content | Style | Content | Style |
| 2B-121 | 25.5% | 9.4% | 33.2% | 14.0% | -3.2% | 0.3% | 18.5% | 7.9% |
| 2B-111 | **36.2%** | 8.4% | 39.2% | **15.0%** | 2.1% | 6.7% | 25.8% | 10.0% |
| 2B-122 | 34.2% | 5.5% | 37.5% | 12.4% | 3.6% | **9.3%** | 25.1% | 9.1% |
| 2B-212 | 29.4% | 5.4% | 42.9% | 13.0% | 0.0% | 8.2% | 24.1% | 8.9% |
| 2B-221 | 35.9% | **11.6%** | **44.0%** | 13.5% | **12.6%** | 8.9% | **30.8%** | **11.3%** |
| 2B-102 | 26.7% | 4.5% | 41.5% | 10.9% | 11.1% | -3.6% | 26.4% | 3.9% |
| 8B-111 | 36.9% | 10.2% | **63.1%** | 14.5% | 10.5% | 4.4% | 36.8% | 9.7% |
| 8B-221 | **40.0%** | **16.0%** | 62.8% | **16.1%** | **18.2%** | **11.7%** | **40.3%** | **14.6%** |

Table 21: Ablation results of controllable image captioning on AnyCapEval under different training data ratio configurations. Abbreviations refer to instruction types defined in the main paper.

| Model | Content | | | | | Style | | | | | |
|---|---|---|---|---|---|---|---|---|---|---|---|
| | IPos. | IApp. | Ins. | Per. | Avg. | Brf. | Det. | Thm. | Poe. | Nar. | Avg. |
| InternVL2.5-8B | 1.51 | 3.43 | 4.54 | 2.68 | 3.04 | 2.13 | 1.92 | 1.94 | 2.08 | 2.52 | 2.12 |
| 2B-121 | $2.59_{(+1.08)}$ | $4.35_{(+0.92)}$ | $4.87_{(+0.33)}$ | $3.45_{(+0.77)}$ | $3.82_{(+0.78)}$ | $2.27_{(+0.14)}$ | $1.84_{(-0.08)}$ | $2.27_{(+0.33)}$ | $2.63_{(+0.55)}$ | $2.59_{(+0.07)}$ | $2.32_{(+0.20)}$ |
| 2B-111 | $3.10_{(+1.59)}$ | $4.20_{(+0.77)}$ | $4.78_{(+0.24)}$ | $4.48_{(+1.80)}$ | $4.14_{(+1.10)}$ | $2.20_{(+0.07)}$ | $1.90_{(-0.02)}$ | $2.09_{(+0.15)}$ | $2.80_{(+0.72)}$ | $2.59_{(+0.07)}$ | $2.29_{(+0.18)}$ |
| 2B-122 | $3.14_{(+1.63)}$ | $4.47_{(+1.04)}$ | $4.63_{(+0.09)}$ | $4.08_{(+1.40)}$ | $4.08_{(+1.04)}$ | $2.16_{(+0.03)}$ | $1.61_{(-0.31)}$ | $2.35_{(+0.41)}$ | $2.59_{(+0.51)}$ | $2.54_{(+0.02)}$ | $2.24_{(+0.12)}$ |
| 2B-212 | $2.53_{(+1.02)}$ | $4.31_{(+0.88)}$ | $4.78_{(+0.24)}$ | $4.11_{(+1.43)}$ | $3.93_{(+0.89)}$ | $2.18_{(+0.05)}$ | $1.69_{(-0.23)}$ | $2.06_{(+0.12)}$ | $2.71_{(+0.63)}$ | $2.57_{(+0.05)}$ | $2.23_{(+0.11)}$ |
| 2B-221 | $3.22_{(+1.71)}$ | $4.49_{(+1.06)}$ | $4.82_{(+0.28)}$ | $4.00_{(+1.32)}$ | $4.13_{(+1.09)}$ | $2.27_{(+0.14)}$ | $1.78_{(-0.14)}$ | $2.38_{(+0.44)}$ | $2.71_{(+0.63)}$ | $2.67_{(+0.15)}$ | $2.36_{(+0.24)}$ |
| 2B-102 | $2.62_{(+1.11)}$ | $4.00_{(+0.57)}$ | $4.81_{(+0.27)}$ | $3.99_{(+1.31)}$ | $3.85_{(+0.81)}$ | $2.18_{(+0.05)}$ | $1.65_{(-0.27)}$ | $2.29_{(+0.35)}$ | $2.63_{(+0.55)}$ | $2.35_{(-0.17)}$ | $2.21_{(+0.10)}$ |
| 8B-111 | $3.36_{(+1.85)}$ | $4.34_{(+0.91)}$ | $5.12_{(+0.58)}$ | $3.83_{(+1.15)}$ | $4.16_{(+1.12)}$ | $2.24_{(+0.11)}$ | $2.06_{(+0.14)}$ | $2.15_{(+0.21)}$ | $2.44_{(+0.36)}$ | $2.87_{(+0.35)}$ | $2.33_{(+0.22)}$ |
| 8B-221 | $3.41_{(+1.90)}$ | $4.82_{(+1.39)}$ | $4.91_{(+0.37)}$ | $3.89_{(+1.21)}$ | $4.26_{(+1.22)}$ | $2.33_{(+0.20)}$ | $2.10_{(+0.18)}$ | $2.56_{(+0.62)}$ | $2.59_{(+0.51)}$ | $2.83_{(+0.31)}$ | $2.46_{(+0.34)}$ |

Table 22: Ablation results of Content control for video captioning on AnyCapEval under different training data ratio configurations.

| Model | Content | | | | | | | | |
|---|---|---|---|---|---|---|---|---|---|
| | IPos. | IApp. | IAct. | Ins. | Per. | Mov. | Bkg. | Evt. | Avg. |
| InternVL2.5-8B | 3.08 | 4.33 | 3.44 | 3.78 | 3.26 | 3.16 | 4.56 | 2.55 | 3.52 |
| 2B-121 | $3.44_{(+0.36)}$ | $5.72_{(+1.39)}$ | $3.01_{(-0.43)}$ | $4.62_{(+0.84)}$ | $4.74_{(+1.48)}$ | $6.20_{(+3.04)}$ | $5.43_{(+0.87)}$ | $4.36_{(+1.81)}$ | $4.69_{(+1.17)}$ |
| 2B-111 | $3.51_{(+0.43)}$ | $6.72_{(+2.39)}$ | $3.12_{(-0.32)}$ | $5.03_{(+1.25)}$ | $4.34_{(+1.08)}$ | $6.05_{(+2.89)}$ | $5.81_{(+1.25)}$ | $4.60_{(+2.05)}$ | $4.90_{(+1.38)}$ |
| 2B-122 | $3.95_{(+0.87)}$ | $6.13_{(+1.80)}$ | $2.94_{(-0.50)}$ | $5.05_{(+1.27)}$ | $4.50_{(+1.24)}$ | $6.57_{(+3.41)}$ | $5.37_{(+0.81)}$ | $4.17_{(+1.62)}$ | $4.84_{(+1.32)}$ |
| 2B-212 | $4.30_{(+1.22)}$ | $6.36_{(+2.03)}$ | $3.74_{(+0.30)}$ | $4.79_{(+1.01)}$ | $4.20_{(+0.94)}$ | $6.32_{(+3.16)}$ | $5.84_{(+1.28)}$ | $4.68_{(+2.13)}$ | $5.03_{(+1.51)}$ |
| 2B-221 | $4.67_{(+1.59)}$ | $6.32_{(+1.99)}$ | $3.19_{(-0.25)}$ | $5.22_{(+1.44)}$ | $4.66_{(+1.40)}$ | $6.49_{(+3.33)}$ | $5.49_{(+0.93)}$ | $4.50_{(+1.95)}$ | $5.07_{(+1.55)}$ |
| 2B-102 | $3.60_{(+0.52)}$ | $6.19_{(+1.86)}$ | $5.62_{(+2.18)}$ | $4.80_{(+1.02)}$ | $4.64_{(+1.38)}$ | $5.15_{(+1.99)}$ | $5.15_{(+0.59)}$ | $4.66_{(+2.11)}$ | $4.98_{(+1.46)}$ |
| 8B-111 | $4.88_{(+1.80)}$ | $6.92_{(+2.59)}$ | $6.28_{(+2.84)}$ | $5.39_{(+1.61)}$ | $5.38_{(+2.12)}$ | $5.44_{(+2.28)}$ | $5.98_{(+1.42)}$ | $5.62_{(+3.07)}$ | $5.74_{(+2.22)}$ |
| 8B-221 | $4.95_{(+1.87)}$ | $6.39_{(+2.06)}$ | $7.03_{(+3.59)}$ | $5.49_{(+1.71)}$ | $5.85_{(+2.59)}$ | $5.66_{(+2.50)}$ | $5.49_{(+0.93)}$ | $4.95_{(+2.40)}$ | $5.73_{(+2.21)}$ |

Table 23: Ablation results of Style control for video captioning on AnyCapEval under different training data ratio configurations.

| Model | Style | | | | | |
|---|---|---|---|---|---|---|
| | Brf. | Det. | Thm. | Poe. | Nar. | Avg. |
| InternVL2.5-8B | 1.39 | 1.77 | 2.23 | 1.91 | 2.34 | 1.93 |
| 2B-121 | $1.92_{(+0.53)}$ | $1.88_{(+0.11)}$ | $2.45_{(+0.22)}$ | $2.40_{(+0.49)}$ | $2.39_{(+0.05)}$ | $2.20_{(+0.27)}$ |
| 2B-111 | $1.76_{(+0.37)}$ | $1.77_{(+0.00)}$ | $2.62_{(+0.39)}$ | $2.40_{(+0.49)}$ | $2.61_{(+0.27)}$ | $2.22_{(+0.29)}$ |
| 2B-122 | $1.89_{(+0.50)}$ | $1.80_{(+0.03)}$ | $2.65_{(+0.42)}$ | $2.12_{(+0.21)}$ | $2.42_{(+0.08)}$ | $2.17_{(+0.24)}$ |
| 2B-212 | $1.74_{(+0.35)}$ | $1.77_{(+0.00)}$ | $2.50_{(+0.27)}$ | $2.30_{(+0.39)}$ | $2.68_{(+0.34)}$ | $2.18_{(+0.25)}$ |
| 2B-221 | $1.84_{(+0.45)}$ | $1.80_{(+0.03)}$ | $2.45_{(+0.22)}$ | $2.48_{(+0.57)}$ | $2.39_{(+0.05)}$ | $2.19_{(+0.26)}$ |
| 2B-102 | $1.61_{(+0.22)}$ | $1.88_{(+0.11)}$ | $2.58_{(+0.35)}$ | $2.15_{(+0.24)}$ | $2.58_{(+0.24)}$ | $2.14_{(+0.21)}$ |
| 8B-111 | $1.68_{(+0.29)}$ | $1.93_{(+0.16)}$ | $2.55_{(+0.32)}$ | $2.23_{(+0.32)}$ | $2.74_{(+0.40)}$ | $2.21_{(+0.28)}$ |
| 8B-221 | $2.00_{(+0.61)}$ | $1.88_{(+0.11)}$ | $2.50_{(+0.27)}$ | $2.20_{(+0.29)}$ | $2.61_{(+0.27)}$ | $2.24_{(+0.31)}$ |

Table 24: Ablation results of controllable audio captioning on AnyCapEval under different training data ratio configurations.

| Model | Content | Style | | | |
|---|---|---|---|---|---|
| | Evt. | Brf. | Nar. | Poe. | Avg. |
| GPT-4o | 1.59 | 1.42 | 1.24 | 0.88 | 1.18 |
| 2B-121 | $1.67_{(+0.08)}$ | $1.51_{(+0.09)}$ | $1.38_{(+0.14)}$ | $0.91_{(+0.03)}$ | $1.27_{(+0.09)}$ |
| 2B-111 | $1.82_{(+0.23)}$ | $1.53_{(+0.11)}$ | $1.53_{(+0.29)}$ | $1.00_{(+0.12)}$ | $1.36_{(+0.18)}$ |
| 2B-122 | $1.78_{(+0.19)}$ | $1.53_{(+0.11)}$ | $1.47_{(+0.23)}$ | $0.89_{(+0.01)}$ | $1.30_{(+0.12)}$ |
| 2B-212 | $1.81_{(+0.22)}$ | $1.56_{(+0.14)}$ | $1.45_{(+0.21)}$ | $0.93_{(+0.05)}$ | $1.31_{(+0.13)}$ |
| 2B-221 | $1.89_{(+0.30)}$ | $1.40_{(-0.02)}$ | $1.43_{(+0.19)}$ | $0.96_{(+0.08)}$ | $1.26_{(+0.08)}$ |
| 2B-102 | $1.96_{(+0.37)}$ | $1.37_{(-0.05)}$ | $1.33_{(+0.09)}$ | $0.78_{(-0.10)}$ | $1.16_{(-0.02)}$ |
| 8B-111 | $2.13_{(+0.54)}$ | $1.44_{(+0.02)}$ | $1.40_{(+0.16)}$ | $1.00_{(+0.12)}$ | $1.28_{(+0.10)}$ |
| 8B-221 | $1.73_{(+0.14)}$ | $1.40_{(-0.02)}$ | $1.43_{(+0.19)}$ | $0.96_{(+0.08)}$ | $1.26_{(+0.08)}$ |

Table 25: Instruction-based video captioning on AnyCapEval.

| Model | Pos. | App. | Act. | Ins. | Persp. | Cam. | Bkg. | Evt. | Avg. |
|---|---|---|---|---|---|---|---|---|---|
| *Proprietary Models* | | | | | | | | | |
| GPT-4o | 2.41 | 4.00 | 3.86 | 4.01 | 3.74 | 2.70 | 4.66 | 3.03 | 3.55 |
| +AnyCap-2B | $3.45_{(+1.04)}$ | $6.15_{(+2.15)}$ | $6.26_{(+2.40)}$ | $5.45_{(+1.44)}$ | $4.58_{(+0.84)}$ | $6.53_{(+3.83)}$ | $5.39_{(+0.73)}$ | $4.60_{(+1.57)}$ | $5.30_{(+1.75)}$ |
| +AnyCap-8B | $4.92_{(+2.51)}$ | $6.60_{(+2.60)}$ | $7.68_{(+3.82)}$ | $4.95_{(+0.94)}$ | $5.26_{(+1.52)}$ | $5.67_{(+2.97)}$ | $6.04_{(+1.38)}$ | $4.81_{(+1.78)}$ | $5.74_{(+2.19)}$ |
| *Open-source Models* | | | | | | | | | |
| InternVL2.5-8B | 3.08 | 4.33 | 3.44 | 3.78 | 3.26 | 3.16 | 4.56 | 2.55 | 3.52 |
| +AnyCap-2B | $4.67_{(+1.59)}$ | $6.32_{(+1.99)}$ | $3.19_{(-0.25)}$ | $5.22_{(+1.44)}$ | $4.66_{(+1.40)}$ | $6.49_{(+3.33)}$ | $5.49_{(+0.93)}$ | $4.50_{(+1.95)}$ | $5.07_{(+1.55)}$ |
| +AnyCap-8B | $4.95_{(+1.87)}$ | $6.39_{(+2.06)}$ | $7.03_{(+3.59)}$ | $5.49_{(+1.71)}$ | $5.85_{(+2.59)}$ | $5.66_{(+2.50)}$ | $5.49_{(+0.93)}$ | $4.95_{(+2.40)}$ | $5.73_{(+2.21)}$ |
| Qwen2.5VL-7B | 2.88 | 4.20 | 3.29 | 4.02 | 3.98 | 2.24 | 4.77 | 2.93 | 3.54 |
| +AnyCap-2B | $3.39_{(+0.51)}$ | $5.97_{(+1.77)}$ | $6.23_{(+2.94)}$ | $4.91_{(+0.89)}$ | $5.23_{(+1.25)}$ | $6.30_{(+4.06)}$ | $5.51_{(+0.74)}$ | $4.48_{(+1.55)}$ | $5.25_{(+1.71)}$ |
| +AnyCap-8B | $4.03_{(+1.15)}$ | $6.15_{(+1.95)}$ | $6.58_{(+3.29)}$ | $4.81_{(+0.79)}$ | $6.24_{(+2.26)}$ | $6.04_{(+3.80)}$ | $5.74_{(+0.97)}$ | $4.82_{(+1.89)}$ | $5.55_{(+2.01)}$ |

| Model | Brf. | Det. | Sty. | Poe. | Nar. | Avg. |
|---|---|---|---|---|---|---|
| *Proprietary Models* | | | | | | |
| GPT-4o | 1.47 | 1.52 | 2.77 | 2.52 | 2.48 | 2.15 |
| +AnyCap-2B | $1.97_{(+0.50)}$ | $1.78_{(+0.26)}$ | $2.52_{(-0.25)}$ | $2.60_{(+0.08)}$ | $2.65_{(+0.17)}$ | $2.30_{(+0.15)}$ |
| +AnyCap-8B | $1.95_{(+0.48)}$ | $1.82_{(+0.30)}$ | $2.58_{(-0.19)}$ | $2.50_{(-0.02)}$ | $2.77_{(+0.29)}$ | $2.32_{(+0.17)}$ |
| *Open-source Models* | | | | | | |
| InternVL2.5-8B | 1.39 | 1.77 | 2.23 | 1.91 | 2.34 | 1.93 |
| +AnyCap-2B | $1.84_{(+0.45)}$ | $1.80_{(+0.03)}$ | $2.45_{(+0.22)}$ | $2.48_{(+0.57)}$ | $2.39_{(+0.05)}$ | $2.19_{(+0.26)}$ |
| +AnyCap-8B | $2.00_{(+0.61)}$ | $1.88_{(+0.11)}$ | $2.50_{(+0.27)}$ | $2.20_{(+0.29)}$ | $2.61_{(+0.27)}$ | $2.24_{(+0.31)}$ |
| Qwen2.5VL-7B | 1.63 | 1.65 | 2.55 | 2.40 | 2.29 | 2.10 |
| +AnyCap-2B | $1.66_{(+0.03)}$ | $1.82_{(+0.17)}$ | $2.48_{(-0.07)}$ | $2.30_{(-0.10)}$ | $2.55_{(+0.26)}$ | $2.16_{(+0.06)}$ |
| +AnyCap-8B | $1.92_{(+0.29)}$ | $1.98_{(+0.33)}$ | $2.65_{(+0.10)}$ | $2.45_{(+0.05)}$ | $2.55_{(+0.26)}$ | $2.31_{(+0.21)}$ |

Table 26: Performance of additional models on the image modality of AnyCapEval.

| Model | IPos.↑ | IApp.↑ | Ins.↑ | Per.↑ | Avg.↑ | Brf.↑ | Det.↑ | Thm.↑ | Poe.↑ | Nar.↑ | Avg.↑ |
|---|---|---|---|---|---|---|---|---|---|---|---|
| LLaVA-7B | 1.50 | 2.99 | 3.85 | 2.63 | 2.74 | 1.27 | 0.98 | 1.21 | 0.71 | 1.67 | 1.19 |
| +AnyCap-8B | 3.26 | 4.72 | 5.23 | 4.56 | 4.44 | 2.27 | 1.98 | 2.56 | 2.90 | 2.74 | 2.45 |
| MiniCPM-o | 1.96 | 3.80 | 4.90 | 3.07 | 3.43 | 2.11 | 2.06 | 2.15 | 1.74 | 2.54 | 2.12 |
| +AnyCap-8B | 2.31 | 4.40 | 5.00 | 2.98 | 3.67 | 2.36 | 2.14 | 2.73 | 2.33 | 2.74 | 2.44 |
| YiVL-34B | 1.38 | 1.85 | 1.74 | 1.73 | 1.67 | 0.91 | 0.59 | 0.85 | 0.55 | 1.04 | 0.81 |
| +AnyCap-8B | 3.54 | 4.69 | 4.93 | 4.66 | 4.45 | 1.98 | 2.04 | 2.59 | 2.65 | 2.59 | 2.30 |
| InternVL-38B | 1.69 | 3.79 | 4.92 | 3.01 | 3.35 | 2.15 | 2.02 | 2.17 | 2.06 | 2.64 | 2.21 |
| +AnyCap-8B | 3.31 | 4.90 | 5.00 | 4.29 | 4.38 | 2.38 | 2.15 | 2.72 | 2.63 | 2.82 | 2.54 |

Table 27: Performance of additional models on the video modality of AnyCapEval. Top: Content metrics; Bottom: Style metrics.

| Model | Content | | | | | | | | |
|---|---|---|---|---|---|---|---|---|---|
| | IPos.↑ | IApp.↑ | Act.↑ | Ins.↑ | Pes.↑ | Mov.↑ | Bkg.↑ | Evt.↑ | Avg.↑ |
| Tarsier2-7B | 2.50 | 3.39 | 3.80 | 3.95 | 2.49 | 1.10 | 4.26 | 4.18 | 2.83 |
| +AnyCap-8B | 4.85 | 7.79 | 7.96 | 5.94 | 5.77 | 5.48 | 6.10 | 5.57 | 6.18 |
| LLaVA-Video | 2.77 | 5.89 | 2.84 | 3.67 | 3.10 | 1.34 | 6.15 | 2.65 | 3.55 |
| +AnyCap-8B | 4.37 | 7.08 | 6.20 | 5.68 | 5.79 | 5.35 | 6.62 | 4.35 | 5.68 |
| MiniCPM-o | 2.68 | 3.59 | 3.31 | 4.21 | 4.16 | 1.76 | 4.62 | 1.76 | 3.26 |
| +AnyCap-8B | 4.65 | 5.70 | 6.05 | 5.20 | 5.48 | 5.06 | 6.01 | 5.06 | 5.40 |

| Model | Style | | | | | |
|---|---|---|---|---|---|---|
| | Brf.↑ | Det.↑ | Thm.↑ | Poe.↑ | Nar.↑ | Avg.↑ |
| Tarsier2-7B | 0.63 | 2.08 | 2.12 | 0.42 | 2.45 | 1.51 |
| +AnyCap-8B | 1.87 | 1.90 | 2.65 | 2.38 | 2.55 | 2.26 |
| LLaVA-Video | 1.61 | 1.19 | 3.36 | 2.01 | 2.70 | 2.13 |
| +AnyCap-8B | 1.97 | 2.11 | 2.74 | 0.99 | 1.32 | 2.23 |
| MiniCPM-o | 1.29 | 1.40 | 2.30 | 1.73 | 2.48 | 1.84 |
| +AnyCap-8B | 1.97 | 2.11 | 2.74 | 0.99 | 1.32 | 2.23 |

**Instruction**: Craft a detailed description for the image that avoids mentioning the motorcycle.

In the foreground, there is a **grassy area where several people are gathered**. They appear to be engaged in conversation or observing something off-camera. Behind them, there is **a large tree providing** shade over part of the scene. Further back, there is **a building with white siding and a dark roof**. The environment suggests a rural or suburban setting with open space and greenery surrounding the area.

**Instruction**: Describe the location of the trees relative to the clock tower.

**Trees flank both sides of the clock tower**, with one tree **prominently visible on the left side** of the image. This tree extends upward, partially obscuring the lower part of the clock face. Another tree is **situated further back**, closer to the right edge of the image, adding depth to the scene.

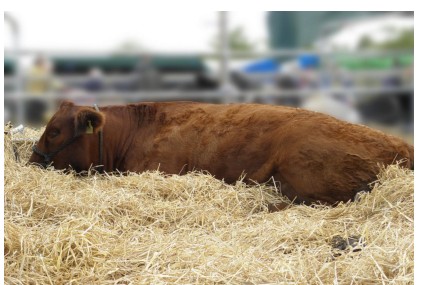

**Instruction**: Please describe the location of other visible elements in the background.

**In the background**, there are several indistinct figures that appear to be people standing or moving around. **These figures are located at varying distances from the cow**, mostly concentrated towards the upper part of the image. There is also a structure that **resembles a tent or canopy, situated further back and slightly to the right side of the image**. The overall setting suggests an outdoor event or gathering area.

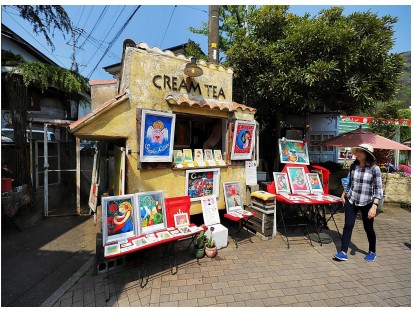

**Instruction**: Describe what can be seen from the perspective of someone standing inside the "CREAM TEA" shop looking outwards towards the street.

**From inside the "CREAM TEA" shop**, one can observe the vibrant display of artwork outside. **The colorful paintings are arranged neatly on tables and easels**, drawing attention with their bright hues and intricate designs. **Outside, there's a person walking by, adding life to the scene**. The shop's interior is partially visible through the open door, showing some items for sale.

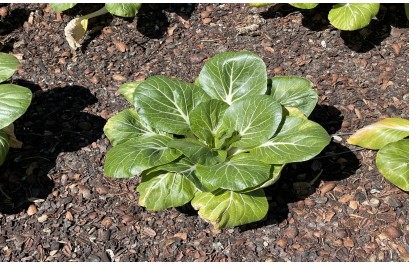

**Instruction**: Describe the central plant in terms of its appearance and notable qualities.

The central plant is **a vibrant green leafy vegetable**, likely a type of lettuce or cabbage. It has **broad, glossy leaves with prominent white veins running through them**. The leaves are slightly crinkled and appear fresh and healthy. The plant **stands out against the dark mulch surrounding it**, indicating it's well-nourished and thriving in its environment.

Figure 7: Qualitative examples of controllable caption generation from images.

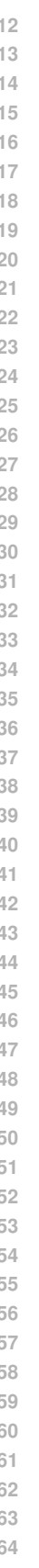

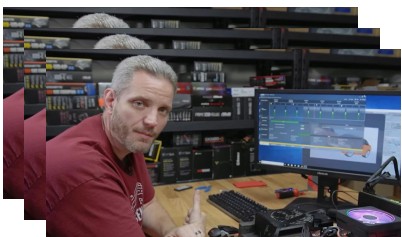

**Instruction**: Describe the actions and movements of the person in the video.

The person is seated at a desk, **gesturing with their hands while speaking**. They occasionally **point towards the computer monitor** displaying graphics. Their **hand movements are expressive** as they explain or discuss something related to the content on the screen.

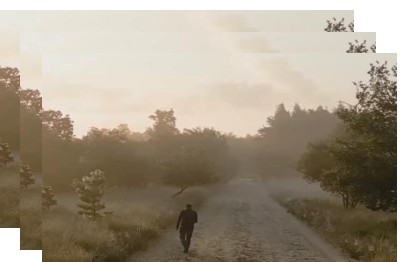

**Instruction**: Describe everything happening in this video, except for the walking motion.

The video captures a **tranquil outdoor setting** during what appears to be early morning or late afternoon. A dirt path stretches **into the distance, flanked by tall grasses and various types of vegetation. Trees line both sides of the path, their leaves bathed in soft sunlight** that filters through them. The atmosphere is calm and serene, with no visible movement apart from the gentle sway of the grasses.

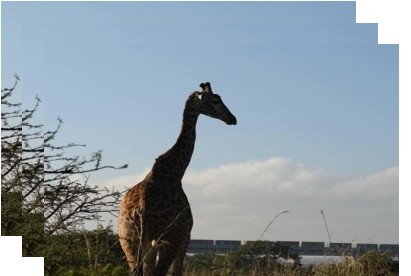

**Instruction**: Describe the appearance and characteristics of the giraffe in the video.

**The giraffe is standing tall with a long neck and legs**. It has **a patterned coat consisting of large, irregular brown patches** separated by lighter lines. Its **head features two small ossicones** on top, and it appears to be looking slightly to the side. The giraffe's body is slender, and its tail is visible at the end.

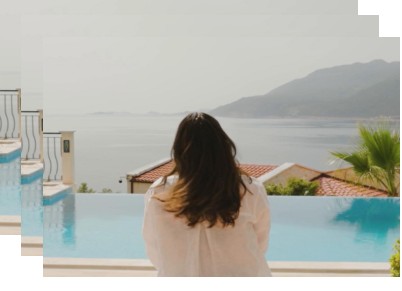

**Instruction**: Please outline the chronological order of events captured in this video.

The sequence **begins with a person sitting by an infinity pool**, facing away from the camera. They are **wearing a white shirt and have long hair that appears to be gently moving due to wind. Throughout the entire sequence, there are no visible actions or changes in their position**; they remain seated on the edge of the pool, gazing out towards the horizon where mountains and water meet.

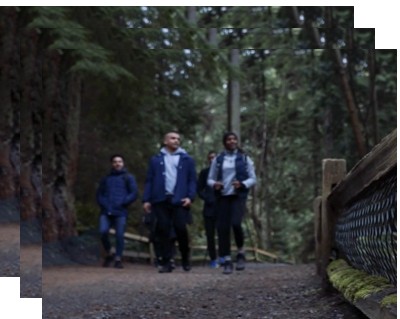

**Instruction**: Tell me about the view from the person's perspective at the back of the group.

The person can see **several people walking forward**. They are **moving along a gravel path bordered by wooden railings covered with moss.**

Figure 8: Qualitative examples of controllable caption generation from videos.

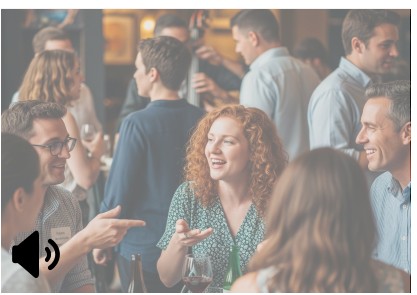

**Instruction**: Provide a description of the events presented in this audio.

The audio captures the sound of a **lively conversation, possibly in a social or public setting**. You can hear **multiple people talking and interacting**, although individual conversations are not easily discernible due to the overlapping voices. There is a general **atmosphere of activity and engagement**, and it's indicative of people interacting with each other in an energetic environment.

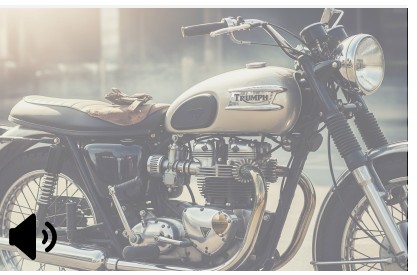

**Instruction**: Describe the events in this audio.

The audio captures the **sound of a vehicle, likely a car or motorcycle, revving its engine**. The **engine's noise increases in intensity**, indicating acceleration or rapid revving. The **primary event here is the running and revving of a motor vehicle engine.**

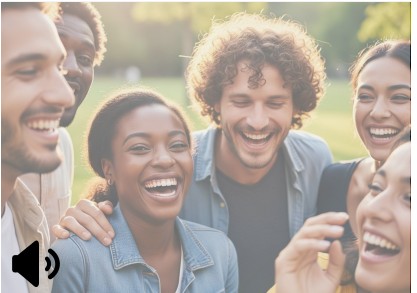

**Instruction**: Create a detailed and comprehensive description of the audio's content.

The audio features a group of **people laughing**. The laughter is **hearty and contagious**, suggesting a joyful and relaxed atmosphere. The sound of laughter indicates that the people are enjoying themselves, possibly sharing a humorous moment or engaging in a lighthearted conversation. The **overall mood conveyed by the laughter is one of happiness and camaraderie**.

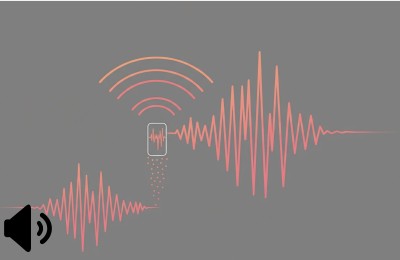

**Instruction**: Please provide a short description of the key information in the audio.

**The audio features a repetitive beeping sound, likely from an alarm or notification system.**

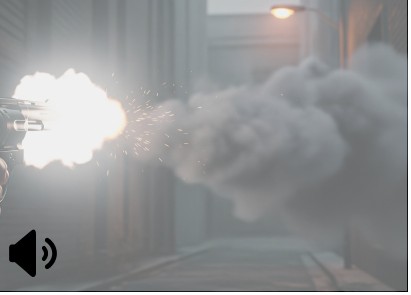

**Instruction**: Please write a brief description of this audio.

**The audio features the sound of a gunshot.**

Figure 9: Qualitative examples of controllable caption generation from audios.

Table 28: Video generation evaluation. Metrics: Visual Quality (VQ), Temporal Consistency (TC), Dynamic Degree (DD), Text–Video Alignment (TVA), and Factual Consistency (FC).

| Model | VQ | TC | DD | TVA | FC | Avg |
|---|---|---|---|---|---|---|
| Wan2.1-T2V-14B | 2.84 | 2.78 | 2.66 | 2.76 | 2.77 | 2.74 |
| +AnyCap-8B | 3.19 (+0.35) | 3.06 (+0.28) | 3.18 (+0.52) | 3.07 (+0.31) | 3.00 (+0.23) | 3.10 (+0.36) |
| HunyuanVideo | 2.98 | 2.89 | 2.94 | 2.90 | 2.83 | 2.91 |
| +AnyCap-8B | 3.61 (+0.63) | 3.48 (+0.59) | 3.62 (+0.68) | 3.34 (+0.44) | 3.46 (+0.63) | 3.50 (+0.59) |

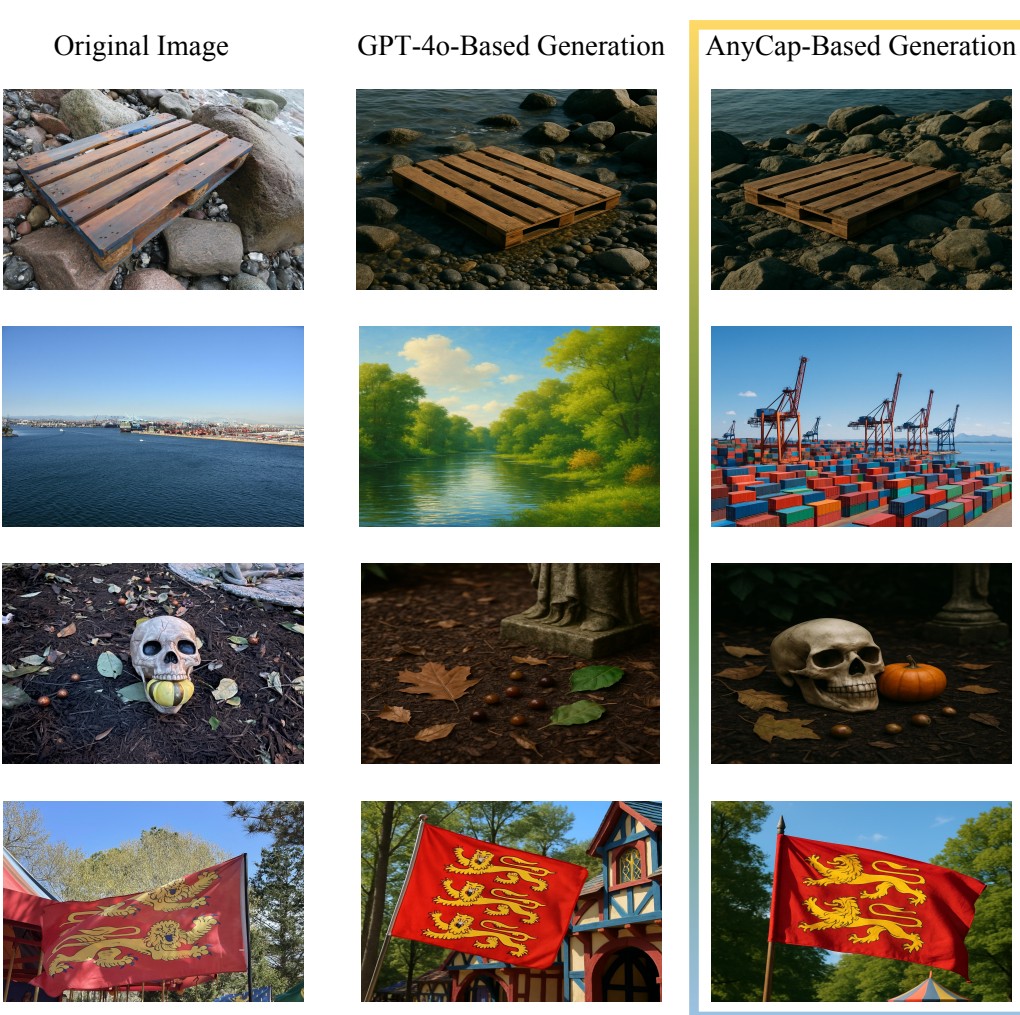

Original Image     GPT-4o-Based Generation     AnyCap-Based Generation

Figure 10: Images in the first column are original real images. The second column shows images generated from GPT-4o captions using DALL·E 3, while the third column shows results from our refined captions. Our method leads to more faithful visual content and better alignment with the original image semantics.

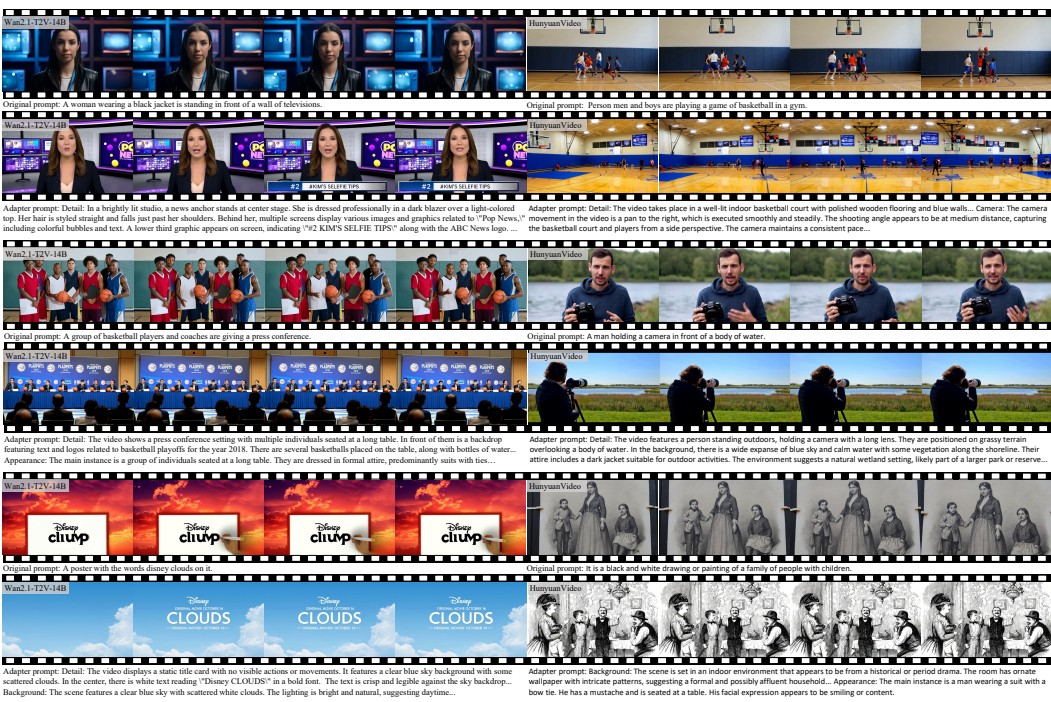

Figure 11: Enhanced text-to-video generation through refined caption quality. Videos in the top row are generated from original dataset captions. The bottom row showcases videos generated using our model's refined captions, demonstrating improved visual fidelity and more expressive camera motion.

