# OpenReview forum: "AnyCap: Omni-Modal Captioning with Instruction Alignment"
_ICLR.cc/2026/Conference — ICLR 2026 Conference Withdrawn Submission_

### Official Review · Reviewer_cbZG · 2025-10-22

**Soundness:** 2
**Presentation:** 3
**Contribution:** 2
**Rating:** 2
**Confidence:** 5

**Summary:**

This paper presents a comprehensive framework named AnyCap for omni-modal (image, video, audio) instruction-aligned captioning. The authors identify key challenges in this domain, including the weak instruction-following ability of existing models, the difficulty of omni-modal generalization, and the lack of dedicated data and evaluation metrics. To address these issues, the paper makes three main contributions:

AnyCap Framework: A plug-and-play module that refines captions from frozen base Multimodal Large Language Models (MLLMs) using a residual-correction mechanism to align them with user instructions.

AnyCapData: A new, large-scale dataset of ~300k samples across three modalities, featuring 28 fine-grained instruction types and structured as (instruction, preferred_caption, less_preferred_caption) triplets.

AnyCapEval: A new evaluation benchmark that decouples assessment into content (measured by a proposed Keypoint Density metric, KPD) and style (measured using a rubric-guided MLLM-as-judge).

The authors conduct extensive experiments showing that AnyCap significantly improves the instruction-following capabilities of various powerful MLLMs like GPT-4o and InternVL across all three modalities.

**Strengths:**

The paper focuses on a practical and important problem: as MLLMs become more widespread, the ability to control their outputs via natural language instructions is crucial for many downstream applications. The omni-modal scope (image, video, and audio) is ambitious and aligns with the future direction of multimodal AI. The authors have invested a significant amount of effort in building a complete ecosystem around the problem, including the model, a large-scale dataset, and a tailored evaluation benchmark. The experiments are extensive, covering multiple state-of-the-art open-source and proprietary base models, and demonstrating consistent and substantial improvements. The empirical results are strong and convincing. The paper is well-written, clearly structured, and easy to follow.

**Weaknesses:**

Despite the impressive engineering effort and strong empirical results, the paper's primary weakness lies in its limited academic novelty across its three main contributions. The work feels more like a skillful application and scaling of existing ideas rather than a fundamental research advancement.

**Lack of Methodological Originality in AnyCap Framework**: The core training paradigm of AnyCap—using a smaller model to learn a residual correction over the output of a larger, frozen model—is heavily inspired by, if not identical to, the method proposed in Aligner, which the paper cites. While the authors apply this idea to a new, multimodal context, the paper does not sufficiently articulate the novel challenges or methodological adaptations required for this extension. The contribution appears to be an application rather than an invention. Besides, there exist some previous works [1] focusing on image caption refining as well.

**Incremental Novelty in Data Construction**: The practice of using powerful proprietary models (like the GPT series) to synthesize large-scale, instruction-rich datasets for training smaller models is now a well-established paradigm in the field. Previous works [2] have already demonstrated the effectiveness of using strong multimodal models to generate detailed captions. While the scale, omni-modal nature, and diversity of instructions in AnyCapData are commendable, the underlying methodology for data creation does not introduce a new technique.

**Limited Innovation in Evaluation**: The use of an MLLM-as-a-judge is also a well-explored research direction for evaluating generative models, with established benchmarks and analyses [3]. Decoupling evaluation into "content" and "style" is a logical refinement, and the proposed KPD metric is a reasonable design choice to penalize verbosity. However, these represent incremental improvements over existing evaluation frameworks rather than a fundamentally new approach to automated assessment.

**Unclear Motivation and Missing Details**: The motivation for why the residual-correction approach is the most suitable for this problem, beyond its empirical success, is not deeply explored. Furthermore, the paper lacks crucial implementation details. It mentions training 2B and 8B variants of AnyCap, but fails to specify the base multimodal model architecture (e.g., LLaVA, Qwen-VL, etc.) used for these variants and the rationale behind this choice. This omission hinders reproducibility and a full understanding of the model's properties.

Generally, the paper presents a powerful and useful system, but its contributions are primarily in engineering and system-building rather than in creating new techniques for the academic community.

[1] Lai, Zhengfeng, et al. "Revisit large-scale image-caption data in pre-training multimodal foundation models." arXiv preprint arXiv:2410.02740 (2024).

[2] Li, Xiaotong, et al. "Densefusion-1m: Merging vision experts for comprehensive multimodal perception." Advances in Neural Information Processing Systems 37 (2024): 18535-18556.

[3] Chen, Dongping, et al. "Mllm-as-a-judge: Assessing multimodal llm-as-a-judge with vision-language benchmark." Forty-first International Conference on Machine Learning. 2024.

**Questions:**

1. The results show significant improvements on average. However, it would be highly insightful to see an analysis of AnyCap's failure modes. Are there specific types of instructions, modalities, or initial captions where the residual correction fails or even degrades the quality? Such an analysis would provide a more nuanced understanding of the method's limitations.

2. The aligner-style training paradigm outperforms both SFT and DPO in the experiments. How does it compare with GRPO?

---

### Official Review · Reviewer_Egxr · 2025-10-23

**Soundness:** 2
**Presentation:** 3
**Contribution:** 2
**Rating:** 4
**Confidence:** 3

**Summary:**

This paper proposes AnyCap, an omni-modal captioning framework capable of handling arbitrary numbers and types of image inputs through a unified visual-language alignment module. The model combines a lightweight vision encoder with an adaptive alignment layer and a language generator (e.g., Qwen2.5VL). Experiments across several visual domains demonstrate improved generalization and caption quality over existing multimodal models.

**Strengths:**

1. The paper introduces a clearly defined and practically meaningful task: omni-modal captioning.
2. The proposed architecture is modular and can be easily integrated with various LLM-based captioners.
3. Comprehensive experiments show consistent improvements across standard captioning benchmarks.
4. The inclusion of a diverse 300K-sample dataset strengthens cross-domain generalization.

**Weaknesses:**

1. The reviewer has some concerns regarding the parameter fairness. The paper reports results using configurations such as Qwen2.5VL-7B + AnyCap-8B, which raises ambiguity about the total parameter scale. If both modules are active during inference, the effective model size approaches 15B parameters, making direct comparison to 7B/8B baselines potentially unfair. It is suggested that the authors provide more clarifications, such as FLOPs and memory usage to justify fairness.
2. Although the paper mentions semi-automatic filtering and LLM-assisted validation, the details of dataset generation, annotation quality, and potential biases are under-specified. These are key features to evalute a dataset and it is necessary to provide more clarifications.
3. The computational overhead of the alignment and cross-image attention modules appears nontrivial, but is not quantified in terms of model latency or throughput.
4. While the Omni-Modal Alignment module is central to the contribution, its individual effect is not deeply dissected (e.g., alternative projection functions or loss terms). It is suggested that the authors provide more clarifications or ablation results on the alignment layers.
5. The baselines provided, such as GPT-4o, InternVL2.5-8B, and Qwen2.5VL-7B, are relatively old. It is suggested that the authors provide more combinations with newer large-scale multimodal models to better evaluate the model’s performance.

**Questions:**

Please refer to weaknesses.

---

### Official Review · Reviewer_bYC5 · 2025-10-23

**Soundness:** 3
**Presentation:** 3
**Contribution:** 2
**Rating:** 4
**Confidence:** 4

**Summary:**

This paper tackles omni-modal, instruction-aligned captioning (images, video, audio) and proposes AnyCap, a plug-and-play residual post-editor that corrects base-model captions to follow user instructions without finetuning the base model. The workflow first obtains an initial caption from an existing model, then conditions a lightweight editor on the instruction + modality features + initial caption to produce the aligned output.

**Strengths:**

Plug-and-play residual editor, unified across modalities. A practical, lightweight post-editing module that brings instruction-aligned captioning to images / video / audio without finetuning the base model—preserving base capabilities while improving controllability; easy to attach to existing pipelines.

Well-scoped data resource (AnyCapData). Provides a ~300k (q, c, a) triplet corpus spanning three modalities with 28 instruction types covering both content and style controls—useful breadth for training and future research.

Task-appropriate evaluation (AnyCapEval). Introduces a content vs. style evaluation scheme (keypoint coverage + rubric-guided style scoring) that targets instruction following more directly than generic MT-style metrics, with initial human correlation checks.

**Weaknesses:**

1) The paper claims to be the first to achieve instruction-aligned captioning in an omni-modal setting. However, these works (Hurst et al., 2024; Anil et al., 2023; Xu et al., 2025) appear to pursue similar goals. The authors are encouraged to carefully re-evaluate the use of “the first” and more precisely position their contribution with respect to these concurrent or prior efforts.

2) The current mathematical symbols lack clarity and cognitive intuitiveness. It is recommended to revise the notation for better readability.

3) What is the core technical difference between AnyCap and Aligner? If the primary distinction is that AnyCap incorporates the original input beyond Aligner, this may appear incremental.

4) The paper mentions that 5% of the data was sampled for human verification of AnyCapData. Is this sample size sufficient to ensure data quality and representativeness?
5) In the experiments, the authors rely on self-constructed metrics to evaluate the proposed model and verify its effectiveness. However, the use of custom, non-standard metrics without validation against established baselines may undermine the credibility and generalizability of the results.

**Questions:**

1）The authors argue that other methods rely on direct LLM-based scoring and suffer from high randomness across instruction types. However, the proposed AnyCapEval also employs GPT-4o as the evaluator. Could the authors clarify whether this introduces a similar source of randomness?

2) In the data construction pipeline, why are “descriptions” (Table 1) not used directly as user instructions? Instead, the authors employ MLLMs to generate instructions.

3) For content evaluation, why not leverage existing metrics instead of proposing a new one? Since content evaluation is a fundamental aspect of captioning tasks, several well-validated evaluation metrics are already available.

4) In the content evaluation, how are the keypoint sets defined and annotated? Is this process conducted via human annotation?

---

### Official Review · Reviewer_wkbm · 2025-10-29

**Soundness:** 2
**Presentation:** 2
**Contribution:** 3
**Rating:** 4
**Confidence:** 4

**Summary:**

This paper presents AnyCap, a plug-and-play framework designed to bring instruction alignment to omni-modal (image, video, audio) captioning tasks. The core technical strategy is a residual-correction paradigm: AnyCap refines captions generated by strong, but inherently uncontrolled, base models into outputs that explicitly follow user-provided natural language instructions, without modifying the original model weights. To support this, the authors introduce AnyCapData, a large-scale triplet-based dataset for instruction-aligned captioning across three modalities, and propose the AnyCapEval evaluation benchmark, which decouples content and style aspects for fine-grained assessment. Extensive experiments demonstrate AnyCap's effectiveness in improving both content and style instruction-following across a range of modalities, models, and benchmarks.

**Strengths:**

1. Practical Residual Correction Framework: The residual-correction approach—refining base captions conditioned on both modalities and user instructions—offers a low-cost, flexible pathway to enhance controllability for diverse captioners, including API-based/closed models, without needing to retrain or fine-tune large base models. This plug-and-play property is valuable in fast-evolving multi-modal settings.
2. Unified Omni-modal Design: AnyCap processes images, videos, and audio with a single architecture, using shared embeddings with modality-specific encoders and projectors (as shown in Figure 2b). This unified approach is well-motivated given the growing need for universal captioning models.
3. Data and Evaluation Resources: The construction of AnyCapData, with 28 detailed instruction types and attention to both modality and instruction diversity (see Table 1), is a meaningful contribution. The triplet annotation aligns well with the correction objective, and the data collection pipeline shows evidence of both scale and attempted quality control.

**Weaknesses:**

1. Lack of Sufficient Novelty in Correction Paradigm: The central residual-correction idea heavily builds on the precedent set by Aligner (Ji et al., 2024). While AnyCap generalizes this to multi-modality, many of the implementation details—base model freezing, reference-based correction, triplet construction—are conceptually close to prior text-correcting systems. The creative leap may not go far enough in algorithmic innovation for a top-tier venue, despite practical value.
2. Evaluation Relies Heavily on LLM-Based Metrics: Although AnyCapEval attempts to lower evaluation variance and improve reliability (see Figure 3b and rubric design), it is fundamentally centered on GPT-4o judgments (in both KPD matching and style grading), with ground-truths and keypoints at the mercy of LLM outputs. Despite a small human study, concerns about LLM bias, overfitting to LLM preferences (especially with AnyCap optimized on LLM-generated corrections/instructions), and reproducibility remain. For diversity and transparency, inclusion of purely human evaluation or strong non-LLM baselines across all major experiments would be important.

**Questions:**

1. How robust is AnyCap to weak, biased, or out-of-domain base captions? Do performance improvements persist when paired with much weaker, outdated, or non-English base models? Are there ablation studies on failure or boundary conditions?
2. What steps were taken to ensure diversity and true instruction alignment in the AnyCapData dataset, beyond the limited manual checks? Have you measured style/content diversity, LLM bias, or evidence of repetitive patterns in auto-generated pairs?
3. In light of the reliance on LLM-based scoring in AnyCapEval, can you provide further breakdowns of human vs. LLM disagreement (beyond aggregate rates shown in Figure 3b)? Are there identifiable domains or instruction types where the automatic evaluation fails or disagrees with human annotators?

---

### Author Response · Authors · 2025-12-03

We thank all reviewers for their constructive feedback and for recognizing the practical value and broad applicability of **AnyCap** as a unified, plug-and-play framework for omni-modal instruction-aligned captioning.

## On novelty and contribution

We respectfully point out that AnyCap goes beyond previous text-only 3H-style(Honest, Helpful, Harmless) correction approaches in several important aspects:

1. It introduces a **unified residual-correction paradigm** across three modalities (image, video, and audio).
2. It incorporates **explicit, data-specific control instructions** into the input.

Together, these enable alignment with human intent along **three dimensions**—multimodal information, semantic content, and control-instruction signals—rather than performing purely semantic correction as in 3H.

## On evaluation and reproducibility

As explained in §2 and Appendix B.1, **rule-based metrics** fail almost entirely in captioning—especially in controllable captioning—motivating the use of **LLM-based scoring**. We further designed a **judgment-based (rather than generation-based)** evaluation protocol to minimize randomness. Additionally, extensive human studies (Fig. 3(b), Fig. 5, §4.4) demonstrate strong consistency between human and LLM assessments.

## On comments regarding missing details

We regret that some reviewers felt certain descriptions were missing, **although these points are discussed prominently in the main paper and supplementary material**. The overlooked information includes (but is not limited to):

- scores on existing public benchmarks,
- details of the data-generation pipeline (Appendix),
- performance of additional model variants (Appendix),
- the instruction taxonomy table (treated as user instructions),
- discussion of limitations of existing metrics, and
- the human evaluation procedure.

## On Reviewer cbZG’s comments

We respectfully disagree with this reviewer’s assessment, as it appears neither fully familiar with the domain nor entirely balanced.

- The internal reasoning is inconsistent: the review acknowledges strong system design, robust experiments, and clear empirical gains in the *Strengths* section, yet assigns a “Rating = 2 (reject)” based chiefly on perceived incrementality.
- The criticism of reproducibility is misplaced: all base-model architectural details are already provided in the supplementary material.
- The request for additional rule-based training comparisons (e.g., GRPO) is unreasonable given that rule-based metrics are shown to be ineffective in this task setting.
- Finally, the use of extreme expressions such as “identical to” or “scaling of existing ideas” overlooks our substantial extensions in omni-modal scope and alignment dimensions.

---

### Note · Authors · 2026-01-06

I have read and agree with the venue's withdrawal policy on behalf of myself and my co-authors.